# ON THE QUANTIZATION OF NEURAL VIDEO CODECS

## ABSTRACT

Full-precision floating-point neural image and video codecs pose significant challenges in power consumption, storage requirements, and cross-platform interoperability, particularly when deployed on resource-constrained devices. To address these issues, network quantization techniques have been extensively studied for neural image codecs. However, the quantization of neural video codecs remains largely unexplored. Unlike quantizing neural image codecs, quantizing neural videos codecs requires significantly more effort. Many coding components operate on temporally correlated data and often rely on features propagated from previous frames, introducing additional sensitivity to both cross-platform round-off errors and network quantization. This work presents the first systematic and algorithmic study of quantization effects across multiple neural video coding frameworks and temporal buffering strategies. Extensive analyzes are conducted to evaluate how various combinations of coding frameworks and temporal buffering strategies respond to various quantization schemes in terms of coding performance and computational complexity. This work offers actionable insights into the future development of neural video codecs.

## 1 INTRODUCTION

State-of-the-art neural video codecs (NVCs) have demonstrated superior performance than traditional codecs (e.g. HEVC/H.265 and VVC/H.266). The coding gains, however, come at the cost of increased model size, higher computational complexity, larger temporal buffers, and other practicality issues. The development of NVCs is entering a critical stage in which the focus shifts from maximizing coding performance to addressing practicality challenges. The first challenge is cross-platform interoperability (Ballé et al., 2019). Like neural image codecs, neural video codecs typically rely on floating-point arithmetic, which can produce platform-dependent variations in output. Temporal drifting errors may thus arise when encoding and decoding occur on distinct platforms. This is because even minor numerical perturbations in each decoded video frame can propagate and accumulate over time via inter-frame prediction, leading to significant quality degradation in the decoded video, as illustrated in Fig. 1a. The second challenge is the high memory access overhead. Fig. 1b depicts a typical implementation of video codecs, traditional or neural, on resource-limited devices. It adopts an on-chip, block/patch-based processing pipeline while storing temporally propagated information for motion compensation and inter-frame prediction in off-chip memory. Most neural video codecs involve frequent access to such a temporal buffer, making memory operations highly energy-intensive. Power dissipation can become prohibitively high (Perleberg et al., 2024). Although intermediate features in a neural network-based autoencoder can impose high memory access demands, this issue can be largely mitigated by the block/patch-based processing pipeline (Fig. 1b). By decomposing intermediate features into blocks/patches for on-chip processing, this approach eliminates the need for their off-chip storage.

One of the most effective ways to address these challenges is by applying neural network quantization. Quantization in this context refers to representing the weights and activations of neural networks, originally in 32-bit floating-point precision (FP32), using lower bit-width formats, such as 16-bit integers (INT16) (Kuzmin et al., 2022; Nagel et al., 2021; Jacob et al., 2018; Gholami et al., 2021). There are two primary approaches to quantizing neural networks: post-training quantization (PTQ) and quantization-aware training (QAT). When proprietary training data and recipes are unavailable, PTQ applies quantization to a pretrained model without modifying the training process, offering simplicity and low computational cost, though sometimes at the expense of the model's capability. In contrast, QAT incorporates quantization into the training process, allowing the model

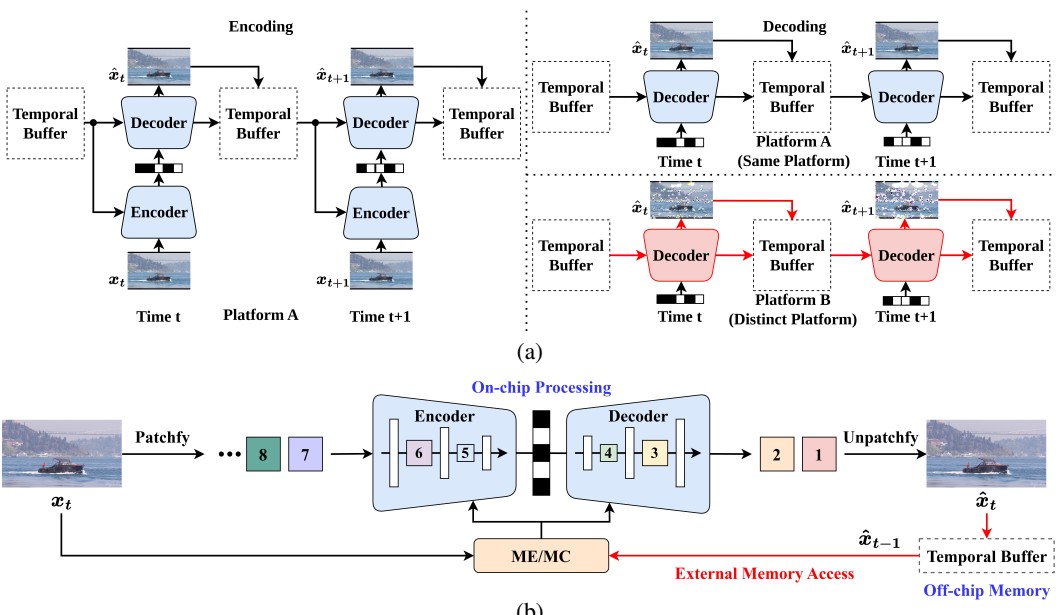

Figure 1: (a) Comparison of video encoding and decoding on the same versus different platforms. **Top-right:** decoding is conducted on the same platform as encoding. **Bottom-right:** decoding is conducted on a different platform than encoding. (b) Patch-wise processing pipeline.

to learn and compensate for quantization-induced errors. QAT typically yields better performance than PTQ, especially with low bit-widths (Kuzmin et al., 2022).

Unlike quantizing neural image codecs, quantizing neural videos codecs requires significantly more effort due to inter-frame dependencies. Typically, video codecs not only reconstruct individual frames, but also model motion and temporal dependencies across frames. As such, the encoding/decoding components operate on temporally correlated data and often rely on features propagated from previous frames, which introduces additional sensitivity to both cross-platform round-off errors and network quantization. Consequently, errors introduced in earlier stages of decoding can propagate and accumulate over time, leading to severe quality degradation Conceição et al. (2025). Therefore, it is essential to also quantize the variational autoencoder (VAE) decoding path to ensure consistency and preserve reconstruction quality, beyond just quantizing the hyperprior decoding path as is common in neural image codecs. Quantization offers several advantages for learned video codecs. On the one hand, it reduces the computational complexity of operations, lowers memory access demands, and promotes cross-platform consistency by minimizing the effects of floating-point variability. On the other hand, lowering numerical precision can introduce quantization errors that degrade coding performance. This trade-off depends on the choice of quantization bit-width. Higher bit-widths provide greater precision, but lower bit-widths reduce complexity and power requirements. Selecting an appropriate bit-width is essential to balancing between coding efficiency and computational cost.

While quantization has been extensively studied in the context of neural image compression, its application to neural video codecs (Gomes et al., 2025) remains largely underexplored. There is limited understanding of how to effectively quantize the diverse and temporally interdependent components in a video decoder. Notably, this work is not restricted to a specific codec design tailored for a specific hardware platform, but rather aims to provide insights that generalize across different coding frameworks and temporal buffering strategies. While each individual component may be implemented in various ways, we adopt those from the state-of-the-art DCVC-FM (Li et al., 2024) in order to derive most meaningful insights into future codec development; moreover, our analysis ensures that various coding frameworks share similar components in order to assess fairly their merits and faults. We experiment with both PTQ and QAT using uniform quantizers, which are broadly supported across modern compute platforms. Specifically, our work has the following unique contributions.

(1) We analyze multiple neural video coding frameworks and temporal buffering strategies to examine how quantization affects their coding performance across a wide range of bit-widths (Sections 4.2

Table 1: Comparison with prior methods.

| Codecs | Coding Frameworks | Temporal Buffers | Component-wise analysis | Mixed Precision PTQ | Sub-16 bit Quantization |
|--------|-------------------|------------------|-------------------------|---------------------|-------------------------|
| MobileNVC | RC | Explicit | × | × | ✓ |
| DCVC-RT | CC | Implicit | × | × | × |
| Ours | CC, MCR | Explicit, Implicit, Hybrid | ✓ | ✓ | ✓ |

and 4.3). (2) We conduct extensive analyses to assess how each decoding component responds to quantization effects (Section 4.2). (3) Based on these analyses, we explore a mixed-precision quantization scheme for various coding variants to strike a balance between coding efficiency and computational complexity (Section 4.4). To the best of our knowledge, this work represents the first systematic study of quantization effects on neural video codecs from a holistic perspective.

## 2 RELATED WORK

Quantization in neural image and video codecs plays a crucial role in reducing memory and power consumption, as well as ensuring cross-platform interoperability, while maintaining high coding performance. Prior works on neural image codecs (Johannes Ballé, 2019; Esin Koyuncu & Kaup, 2022) demonstrate that quantizing the weights and activations of hyperprior components can effectively mitigate decoding errors. In (Koyuncu et al., 2024), it is shown that applying post-training quantization (PTQ) to the entire decoder can achieve bit-exact results for cross-platform operations. However, using uniform precision across all coding components often leads to suboptimal coding performance. Heming Sun (2020) perform quantization on individual weight groups, achieving a 75% reduction in model size compared to the FP32 baseline. Mixed-precision quantization (Hossain et al., 2024) has also been proposed to balance computational complexity and coding efficiency.

Although quantization has been widely explored in neural image codecs, limited research has addressed quantization strategies for different video coding frameworks or the sensitivity of individual decoding components. Most existing approaches apply uniform quantization schemes. For example, MobileNVC (van Rozendaal et al., 2024) applies 8-bit quantization to both weights and activations of the inter-frame encoder and decoder using a multi-stage process—incorporating PTQ and QAT—based on the AIMET framework (Nagel et al., 2021; Siddegowda et al., 2022), with a focus on mobile deployment. However, this approach yields limited coding performance. Conceição et al. (2025) investigated cross-platform round-off errors in the SSF compression framework (Agustsson et al., 2020) and mitigated them by introducing 8-bit quantization in the hyperprior decoder, implemented using the PyTorch quantization framework. A more recent method, DCVC-RT (Jia et al., 2025), quantizes both decoder weights and activations to 16-bit integers (INT16), achieving deterministic inference across platforms. Nevertheless, the impact of lower precision (sub-16-bit) quantization on coding performance across different coding frameworks remains an open issue. Table 1 summarizes the key differences between this and other prior works in terms of their scopes and quantization approaches. RC, CC and MCR refer to Residual Coding, Conditional Coding, and Masked Conditional Residual Coding, respectively (Fig. 2).

## 3 NEURAL VIDEO CODECS

### 3.1 LEARNED VIDEO CODING FRAMEWORKS

The incorporation of temporal correlation in inter-frame codec design has evolved into four mainstream approaches: (1) residual coding (RC), (2) conditional coding (CC), (3) conditional residual coding (CRC), and (4) masked conditional residual coding (MCR). Early learned video codecs (Agustsson et al., 2020; Hu et al., 2021; Lu et al., 2020; Lin et al., 2020; Liu et al., 2020; Lu et al., 2019; Yang et al., 2020) predominantly follow RC (see Fig. 2a), which adopts the same inter-frame coding principle as traditional video codecs. That is, it performs a linear temporal prediction of the current coding frame $x_t$ based on a temporal predictor $x_c$, generated by motion compensating a previously decoded reference frame $\hat{x}_{t-1}$.

Instead of forming a linear prediction of $x_t$, conditional coding (CC) (Ho et al., 2022; Hu et al., 2022; Ladune et al., 2020; Li et al., 2021; 2022; 2023; 2024; Sheng et al., 2023; 2024a;b; Shi et al., 2022) (Fig. 2b) replaces linear temporal prediction with a potentially non-linear approach. It achieves this by providing the (motion-compensated) reference frame or its features as the conditioning signal to the inter-frame encoder/decoder. The way the reference frame is utilized for coding $x_t$ is learned from data, imposing no restrictions on whether the prediction is linear or non-linear. Although

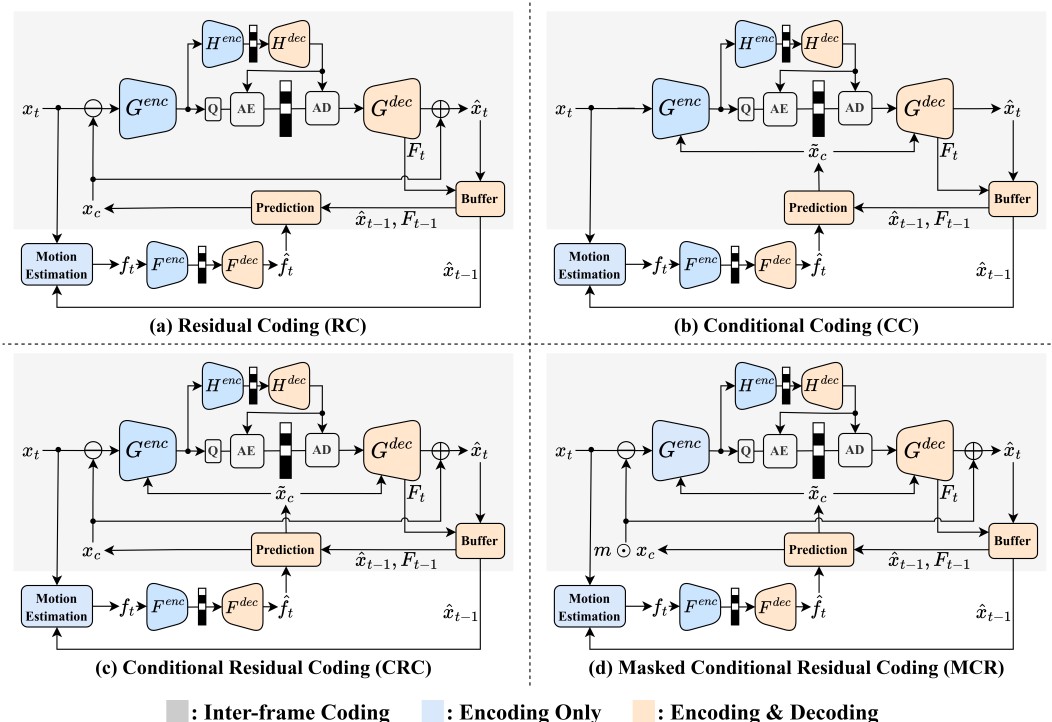

Figure 2: Illustration of neural video coding frameworks, categorized by how temporal correlation is utilized in the inter-frame codec. Each coding framework contains a inter-frame main codec $G^{enc}/G^{dec}$, a hyperprior codec $H^{enc}/H^{dec}$, a motion codec $F^{enc}/F^{dec}$, a motion estimation network, a prediction network, and a temporal buffer. For brevity, some components are omitted. The details of these network architectures are available in the appendix.

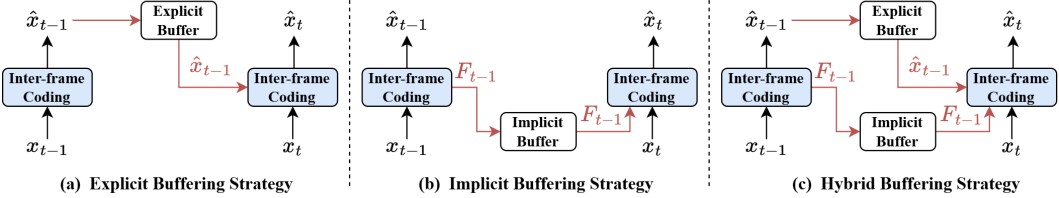

Figure 3: Illustration of temporal buffering strategies for inter-frame coding.

conditional coding (CC) has been adopted by many state-of-the-art NVCs, it is susceptible to the bottleneck issue (Brand et al., 2022).

To overcome the bottleneck issue, CRC (Brand et al., 2024) (see Fig. 2c) introduces a novel approach that blends the strengths of residual coding (RC) and conditional coding (CC). It encodes the prediction residue $x_t - x_c$ using a conditional inter-frame codec that takes $\tilde{x}_c$ as the condition signal. Similar to residual coding (RC), it updates $x_c$ on the decoder side with the decoded residue to reconstruct $x_t$. Furthermore, conditioned on $\tilde{x}_c$, the prediction residue $x_t - x_c$ is coded more efficiently. In Brand et al. (2024), when $x_c$ forms a good prediction of $x_t$, conditional residual coding (CRC) is superior to conditional coding (CC). The coding efficiency of conditional residual coding (CRC) relies heavily on the assumption that $x_c$ provides an accurate prediction of $x_t$. However, this assumption can break down in regions with complex motion or dis-occlusion, causing conditional residual coding (CRC) to perform worse than conditional coding (CC). To mitigate this issue, masked conditional residual coding (MCR) (Chen et al., 2024b) (see Fig. 2d) introduces a soft mask $m$ that adaptively blends CRC and CC at the sub-frame level.

### 3.2 TEMPORAL BUFFERING STRATEGIES

To leverage temporal correlation for coding a video frame $x_t$, learned video codecs store either pixel-domain frames or latent-domain features to assist with subsequent frame coding. In the former

case, the previously decoded frame $\hat{x}_t$ (Agustsson et al., 2020; Brand et al., 2022; 2024; Chen et al., 2024a;b; Ladune et al., 2020; Li et al., 2021; Liu et al., 2020), or even more frames (Lin et al., 2020), is explicitly buffered as a reference frame. In contrast, the latter propagates latent features used for reconstructing $x_t$ (Li et al., 2023; 2024; Sheng et al., 2023; 2024a). Since these latent features are learned completely from training video sequences, there is little control of what information is stored, making their content inherently non-explainable. We term the buffer storing these features as the *implicit* buffer, as opposed to the *explicit* buffer that stores decoded frames.

In Fig. 3, we further illustrate how these buffering strategies differ by conceptualizing them as distinct approaches to constructing recurrent neural networks (RNN) along the temporal dimension for video coding in a rate-distortion sense. To see this, we note that the coding frameworks in Fig. 2 all process input video frame-by-frame. Each input video frame is first transformed by an encoder network into its latent representation and entropy encoded into a bitstream, followed by reconstructing it approximately by a decoder network. As a sequence-to-sequence mapping, the successive inter-frame coding process forms an RNN that encodes video frames one at a time, producing reconstructed frames that closely resemble the originals. At first glance, simply replicating an input sequence as output in an RNN appears trivial. However, in the context of video coding, the challenge lies in minimizing the entropy rate of quantized frame latents while maximizing the reconstruction quality of decoded frames. Encoding input frames individually–akin to intra coding–is feasible yet inefficient; the key to rate-distortion optimized video coding is to leverage and propagate temporal information from past frames to exploit inter-frame correlations effectively.

With this conceptualization of RNN, the *explicit* buffering strategy (Fig. 3a can be viewed as an RNN with only a *output-recurrence* connection. That is, it involves the previously decoded frame as the only source of past information when generating latents for the current coding frame. In contrast, the *implicit* strategy (Fig. 3b) relies solely on a "hidden-to-hidden" connection without the output-recurrence connection. It stores and updates only the latent, multi-channel features as a mechanism for propagating past information. As noted in (Goodfellow et al., 2016), an RNN with only the output-recurrence connection has a limited capacity for signal representation. This limitation arises because the reconstructed frame $\hat{x}_{t-1}$ must not only approximate the corresponding input frame $x_{t-1}$, but also serve as a surrogate summarizing the past information. These dual objectives may conflict. Likewise, an RNN with only the hidden-to-hidden connection is suboptimal as it misses the opportunity to exploit the decoded frames, which often correlate strongly with the current coding frame. Goodfellow et al. (2016) also highlight that combining both output-recurrence and hidden-to-hidden connections in an RNN is theoretically more effective than using either alone, as it integrates the advantages of both—namely, leveraging decoded frames without introducing dual objectives on reconstructed frames. Recently, Chen et al. (2025) investigate a hybrid approach that merges explicit and implicit buffering strategies for inter-frame coding, as illustrated in Fig. 3c. This approach incorporates both the decoded frame $\hat{x}_{t-1}$ and hidden features $F_{t-1}$ to generate latents for the input frame $x_t$. We remark that buffering strategies and coding frameworks are two orthogonal aspects. The four frameworks shown in Fig. 2 can be paired with any of the three buffering strategies in Fig. 3. This work investigates how quantization affects the trade-off between coding performance and computational complexity across various combinations of these frameworks and buffering strategies. The attempt at analyzing the interaction between neural video coding frameworks and temporal buffering strategies under quantization constitutes our novel contributions.

### 3.3 SCOPE AND LIMITATIONS

This work aims for a systematic investigation of quantization effects across diverse neural video coding frameworks and temporal buffering strategies. Our objective is to generate actionable insights that inform the future development of neural video codecs (Chen et al., 2021; Shi et al.). Specifically, we focus on variational autoencoder-based codecs, which differ fundamentally from implicit neural representation-based learned video codecs. Moreover, we incorporate the component designs of DCVC-FM, the current state-of-the-art neural video codec. Additional results with other component designs are provided in the Appendix. To examine quantization effects at the algorithmic level, we follow the established practice in prior studies, such as QLIC (Sun et al., 2022) and MP-PTQ (Yu et al., 2025), and conduct quantization in simulated settings. We report results for both PTQ and QAT using uniform and linear quantization, which are widely supported by existing compute platforms (Nagel et al., 2021; van Rozendaal et al., 2024; Jia et al., 2025). For fair comparison across competing methods, we evaluate reductions in MACs, buffer size, and decoder bit-operations. These metrics are intrinsic to algorithm design and independent of the underlying

compute platform. This ensures a more equitable assessment of competing approaches, free from biases introduced by platform choice or implementation quality.

Our work currently has several limitations. We provide an algorithmic study with results simulated on floating-point devices using uniform quantizers, following (Jia et al., 2025; van Rozendaal et al., 2024). These simulations do not implement actual fixed-point operations, which restricts our ability to validate cross-platform interoperability, assess real hardware deployment, or report runtime and power consumption. Furthermore, since hardware support for codec-specific non-uniform quantization remains limited, such schemes are excluded from our simulations. Nevertheless, by identifying modules within neural video codecs that are less sensitive to precision reduction across various scenarios, our study offers guidance for future research toward low-power, on-chip neural video codecs. A more comprehensive validation of quantization effects on real hardware and integer network implementations for deployment will be pursued in future work.

## 4 QUANTIZING NVCS: A CASE STUDY WITH MCR AND CC

We use masked conditional residual coding (MCR) and conditional coding (CC) as a case study to examine the impact of quantization on neural video codecs. We choose these two coding frameworks because MCR is an emerging, new coding framework, while CC is the current mainstream approach to modern neural video coding. First, we assess the sensitivity of the decoder components to quantization. Our investigation focuses on the decoder; we want to ensure bit-exact video reconstruction across different deployment platforms to avoid temporal drifting errors. Next, we analyze how quantization affects activations and network weights, evaluating their impacts on coding performance degradation. Finally, we examine the influence of quantization on the coding performance of the three temporal buffering strategies for inter-frame coding when integrated with MCR and CC.

### 4.1 EXPERIMENT SETTINGS

**Coding variants and notation:** We consider 2 coding schemes (masked conditional residual coding (MCR) and conditional coding (CC)) along with 3 temporal buffering strategies (Explicit, Implicit and Hybrid). All variants share a similar compression backbone derived from DCVC-FM (Li et al., 2024). Additional implementation details are available in the Appendix.

**Evaluation Metrics:** We assess quantization effects by reporting BD-rate increases relative to a FP32 implementation across four widely used datasets, including UVG (Mercat et al., 2020), MCL-JCV (Wang et al., 2016), HEVC Class B–E (Bossen et al., 2013), and HEVC-RGB (Flynn et al., 2013). Each test sequence is encoded for the first 96 frames with an intra period of 32. Experimental results with an intra-period of -1 are reported in the appendix. Following the common practice, we adopt BT.601 for color space conversion between YUV420 and RGB444, with all tested codecs operating in the RGB domain. Additional results for BT.709, which is used in (Li et al., 2023; 2024), are provided in the Appendix. To avoid padding and ensure fair comparisons, each frame is cropped so that its width and height are multiples of 64. Intra coding is applied at scene cuts across all methods. We report PSNR in the RGB domain and bitrate in bits per pixel (bpp). BD-rate is computed by averaging per-frame PSNR-RGB and bpp across all encoded frames to generate a dataset-specific rate-distortion point. The average BD-rates across all test datasets are reported. Negative and positive BD-rate numbers suggest rate reduction and inflation, respectively. All codecs are trained with the Vimeo90k (Xue et al., 2019) and BVI-DVC (Ma et al., 2021) datasets, following similar training protocols. The training details are in the Appendix.

**Quantization methods:** We conduct experiments using PTQ and QAT, applying both approaches separately at each rate point for weights and activations. PTQ adopts a mean square error (MSE)-based configuration, using the Vimeo90k dataset for calibration. The number of calibration samples follows the standard practice outlined in (Nagel et al., 2021). For QAT finetuning, we adopt both the BVI-DVC (Ma et al., 2021) and Vimeo90k (Xue et al., 2019) datasets. Unless otherwise specified, PTQ is used as the default quantization method. The FP32 codecs are implemented with 32-bit floating-point arithmetic without quantizing weights and activations. To report BD-rates for low-precision codecs, the corresponding FP32 implementations are used as anchors. To facilitate evaluation on floating-point hardware, all results are obtained through quantization simulation (Nagel et al., 2021). Following (Nagel et al., 2021), we employ symmetric per-channel quantization for weights and asymmetric per-tensor quantization for activations.

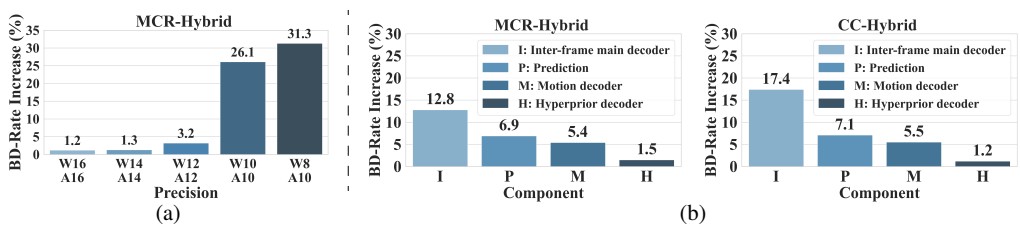

Figure 4: Visualization of the BD-rate increases due to PTQ. (a) Quantizing all decoder components of MCR-Hybrid to sub-16 bit-widths. (b) The component-wise analysis of quantizing MCR-Hybrid and CC-Hybrid to W8A10. The FP32 counterparts serve as anchors.

Table 2: Effects of PTQ on each decoder component in MCR-Hybrid and CC-Hybrid, with the respective FP32 configurations serving as anchors.

(a) Effect of quantizing **activations (A)** for each decoder component.

|  | **MCR-Hybrid** | | | | | **CC-Hybrid** | | | | |
|---|---|---|---|---|---|---|---|---|---|---|
|  | FP32 | W16**A14** | W16**A12** | W16**A10** | W16**A8** | FP32 | W16**A14** | W16**A12** | W16**A10** | W16**A8** |
| **Inter-frame main decoder** |  | 0.2 | 1.5 | 10.1 | - |  | 0.5 | 1.3 | 9.7 | - |
| **Prediction** | 0.0 | 0.0 | 0.1 | 3.0 | - | 0.0 | 0.2 | 0.2 | 1.3 | - |
| **Motion decoder** | (Anchor) | 1.0 | 2.2 | 5.1 | - | (Anchor) | 0.4 | 0.8 | 5.6 | - |
| **Hyperprior decoder** |  | 0.0 | 0.2 | 1.5 | - |  | 0.1 | 0.4 | 1.7 | - |

(b) Effect of quantizing **network weights (W)** for each decoder component.

|  | **MCR-Hybrid** | | | | | **CC-Hybrid** | | | | |
|---|---|---|---|---|---|---|---|---|---|---|
|  | FP32 | **W14**A16 | **W12**A16 | **W10**A16 | **W8**A16 | FP32 | **W14**A16 | **W12**A16 | **W10**A16 | **W8**A16 |
| **Inter-frame main decoder** |  | - | 0.1 | 0.1 | 1.8 |  | - | 0.3 | 0.9 | 6.3 |
| **Prediction** | 0.0 | - | 0.1 | 0.6 | 3.7 | 0.0 | - | 0.1 | 0.4 | 4.2 |
| **Motion decoder** | (Anchor) | - | 1.0 | 1.0 | 1.1 | (Anchor) | - | 0.4 | 0.4 | 0.4 |
| **Hyperprior decoder** |  | - | 0.0 | 0.0 | 0.0 |  | - | 0.1 | 0.1 | 0.5 |

## 4.2 THE SENSITIVITY OF DECODER COMPONENTS TO QUANTIZATION

This section investigates the quantization sensitivity of each decoding component in the challenging scenario W8A10 (i.e. 8 bits for network weights and 10 bits for activations), quantified by BD-rate increases relative to the FP32 anchor (see Section 4.1). The sensitivity is assessed by solely quantizing a single decoding component into W8A10 while the remaining components (in both encoder and decoder) adopt the FP32 format. We choose W8A10 because as shown in Fig. 4a, the MCR-Hybrid variant broke down under W8A10 quantization. To facilitate the analysis, we decompose the decoder into four components: the inter-frame main decoder (I), hyperprior decoder (H), motion decoder (M), and prediction network (P) (which refers collectively to the motion compensation network and condition generation network). Results are reported exclusively for MCR-Hybrid and CC-Hybrid using the 5-channel buffer (B=5), while the results for the other variants (i.e. MCR-Implicit, MCR-Explicit, CC-Implicit, and CC-Explicit) are provided in the appendix. We make the following observations.

**(1) Inter-frame main decoder exhibits the highest sensitivity to quantization.** Fig. 4b shows that in MCR-Hybrid, the inter-frame decoder exhibits the highest sensitivity to quantization, as indicated by its large BD-rate increase, followed by the prediction network. This is because both components play a direct role in reconstructing video frames. As depicted in Fig. 2d, the prediction network generates both the conditional signal $\tilde{x}_c$ for the conditional residual decoder $G^{dec}$ and the temporal predictor $x_c$. Subsequently, $x_c$ is updated by the output of $G^{dec}$, which needs to decode the residual signal via a non-linear synthesis transform. Additionally, quantization errors in the motion decoder lead to motion errors, degrading the quality of $x_c, \tilde{x}_c$ and thereby increasing the bitrate. The hyperprior is the least sensitive to quantization (1.5% BD-rate increase), suggesting that it is amenable to lower precision quantization. CC-Hybrid exhibits a similar trend. However, unlike MCR-Hybrid, the inter-frame main decoder in CC-Hybrid is more severely affected by quantization than the other components. This is because the inter-frame main decoder $G^{dec}$ must reconstruct $x_t$ directly, rather than residual signals as in MCR-Hybrid (see Fig. 2b).

Tables 2a and 2b present an analysis to assess the sensitivity of activations and network weights in each decoder component to quantization. In Table 2a (respectively, 2b), we report BD-rate increases when varying the precision of quantization applied only to the activations (respectively, network weights) of a single decoder component. All remaining components are kept at 32-bit floating point.

Table 3: Effects of PTQ and QAT on decoder components in MCR-Hybrid and CC-Hybrid, with the respective FP32 configurations serving as anchors.

| Precision | Quantization Method | MCR-Hybrid | | | | CC-Hybrid | | | |
|---|---|---|---|---|---|---|---|---|---|
| | | I | P | M | H | I | P | M | H |
| W8A16 | PTQ | 1.8 | 3.7 | 1.1 | 0.0 | 6.3 | 4.2 | 0.4 | 0.5 |
| | QAT | 0.6 | 0.8 | 0.3 | 0.2 | 0.0 | 0.3 | 0.5 | -0.3 |
| W16A10 | PTQ | 10.1 | 3.0 | 5.1 | 1.5 | 9.7 | 1.3 | 5.6 | 1.7 |
| | QAT | 5.6 | 2.6 | 2.5 | 0.0 | 4.7 | 1.4 | 3.2 | 0.1 |

Table 4: The BD-rate comparison of different temporal buffering strategies under PTQ, with the respective Hybrid FP32 configurations serving as anchors.

| Coding Schemes | MCR | | | CC | | |
|---|---|---|---|---|---|---|
| Buffer | Hybrid | Implicit | Explicit | Hybrid | Implicit | Explicit |
| FP32 | 0.0 (Anchor) | 8.3 | 9.3 | 0.0 (Anchor) | 7.9 | 19.1 |
| INT8 | 0.6 | 10.5 | 10.0 | 0.5 | 10.8 | 19.6 |
| BDR (INT8) - BDR (FP32) | 0.6 | 2.2 | 0.7 | 0.5 | 2.9 | 0.5 |

**(2) Activations are more sensitive to quantization than network weights.** This finding is consistent across MCR-Hybrid and CC-Hybrid. A significant coding performance decline is observed when activations, particularly those of the inter-frame main decoder, are quantized to 10 bits, whereas network weights experience a noticeable coding performance drop with 8-bit quantization.

**(3) The inter-frame main decoder in CC-Hybrid requires higher-precision network weights than in MCR-Hybrid.** From Table 2b, 8-bit quantization of network weights results in a significant coding performance loss in CC-Hybrid, whereas MCR-Hybrid remains largely unaffected. This is due to its conditional coding design (see Fig. 2b), where the conditional decoder must directly output the reconstructed frame rather than residual signals as in MCR-Hybrid. Consequently, a higher-capacity inter-frame decoder is required.

**(4) QAT effectively mitigates the performance drop caused by weight quantization.** Table 3 shows that the performance gap introduced by W8A16 quantization under PTQ can be substantially mitigated through QAT. Notably, the inter-frame main decoder exhibits no significant degradation when quantized with QAT. This suggests that the low-precision weights can be effectively adapted to quantized activations, thereby compensating for potential losses in coding performance.

**(5) The activations of the inter-frame main decoder remains critical even with QAT.** Table 3 further indicates that while QAT alleviates the performance loss under W16A10, quantization of the inter-frame main decoder still leads to notable degradation. This confirms that the activations of the inter-frame main decoder require high precision to maintain coding performance.

### 4.3 Impact of Quantization on Temporal Buffering Strategies

This experiment evaluates the impact of temporal buffer quantization on the coding performance of the three buffering strategies: Hybrid, Explicit, and Implicit. In this experiment, all coding components are kept at 32-bit floating-point when the quantization precision for the temporal buffer is varied across six coding variants: MCR-Hybrid, MCR-Implicit, MCR-Explicit, CC-Hybrid, CC-Implicit, and CC-Explicit. From Table 4, we arrive at the following finding:

**(6) Hybrid is superior to Implicit and Explicit in the presence of temporal buffer quantization.** This is evident from the large positive BD-rates of Implicit and Explicit, with Hybrid serving as the anchor. As discussed in Section 3.2, Hybrid employs an RNN structure that incorporates both hidden-to-hidden and output-recurrence connections, theoretically offering greater expressiveness than structures that utilize only hidden-to-hidden (Implicit) or output-recurrence (Explicit) connections. Furthermore, Implicit experiences slightly greater coding loss when the quantization precision is reduced from FP32 to INT8. Table 5 further reports results for two buffer sizes: 5 channels (B=5) and 51 channels (B=51). This experiment is conducted exclusively for Hybrid and Implicit with MCR, in order to see which of the two is more efficient in terms of buffer usage.

**(7) Hybrid (B=5) achieves better coding performance than Implicit (B=51), despite utilizing a smaller buffer size and buffer quantization.** This confirms that Hybrid is more efficient in terms of buffer usage. Notably, Hybrid leverages the prior knowledge that the last reconstructed

Table 5: The BD-rate comparison of MCR-Hybrid and MCR-Implicit with varying buffer sizes (B) after PTQ. "B" denotes the number of channels of full-resolution feature maps for inter-frame coding. The anchor is MCR-Hybrid FP32 with a buffer size of 5.

| Buffering Scheme | Hybrid (B=5) | Hybrid (B=51) | Implicit (B=5) | Implicit (B=51) |
|---|---|---|---|---|
| FP32 | 0.0 (Anchor) | -0.9 | 8.3 | 1.8 |
| INT8 | 0.6 | 0.8 | 10.5 | 3.8 |

Table 6: The complexity and coding performance analysis for MCR-Hybrid B=5 mixed-precision quantization. "w/ MP" denotes mixed-precision quantization with QAT. "w/o MP" is the FP32 model. WPM: Weighted Peak Memory. Refer to Section 4.4 for details on the complexity metrics.

| Setting | Decoder BO (G/pixel) | WPM (Channels) | Buffer size (Channels) | Model Size (MB) | BD-Rate |
|---|---|---|---|---|---|
| w/o MP | 0.89 | 192 | 7.875 | 63.20 | 0.0 (Anchor) |
| w/ MP | 0.12 | 84 | 2.476 | 29.78 | 3.5 |
| Savings | 87% | 51% | 69% | 53% | - |

frame $\hat{x}_{t-1}$ is highly correlated with the current coding frame $x_t$. Therefore, it only needs to learn a few additional latent features for prediction. In contrast, Implicit relies purely on data-driven learning, often resulting in non-compact features with high-precision requirements. With Implicit, a substantial BD-rate gap is observed between the 5-channel and 51-channel configurations, a trend that does not occur in Hybrid.

## 4.4 MIXED-PRECISION QUANTIZATION FOR MCR-HYBRID DECODER

Previous sections have shown that different decoder components exhibit varying degrees of sensitivity to quantization. To achieve a better rate-distortion-complexity, we focus on exploring mixed-precision quantization for the MCR-Hybrid (B=5), assigning higher precision to components that are more sensitive to quantization and lower precision to those that are less sensitive. Specifically, based on our quantization analyses of the activations, weights, and temporal buffer in Tables 2 and Table 4, we identify the minimum precision of activations and network weights precision for each components configurations. We employ QAT to maintain acceptable coding performance of the quantized codecs. The precision settings of individual components under the mixed-precision quantization scheme for the MCR-Hybrid (B=5) and CC-Hybrid (B=5) decoder are provided in the supplementary material. Admittedly, this approach does not account for interactions between decoder components when determining their bit-widths, leaving ample room for further optimization. Even with this ad-hoc strategy, we observe a clear improvement in the complexity-performance trade-off. It is worth noting that our mixed-precision quantization introduces no algorithmic modifications to the codec design and serves as a direct application of our findings. Developing more advanced mixed-precision schemes that automatically determine precision represents a promising direction.

Table 6 presents a comprehensive analysis of its coding efficiency and complexity, contrasting the configuration with mixed-precision quantization (w/ MP) against the full-precision FP32 model (w/o MP). In Table 6, Decoder BO (G/pixel) measures the number of bit operations (Mart van Baalen, 2020) per pixel of decoding components while Weighted Peak Memory quantifies the largest feature size, measured by the number of 32-bit full-resolution feature maps. Buffer Size (Chen et al., 2024b) is expressed as the number of 32-bit full-resolution feature maps stored in the buffer. Model Size refers to the total number of bytes used for network parameters. Decoding time is not reported, since the quantization is simulated in a floating-point computation environment, and the measured decoding time does not reflect the actual reduction achieved by quantization. Further discussion about the complexity is included the appendix. The full resolution refers to the spatial resolution, $W \times H$, of the input video frame. MCR-Hybrid benefits from the MP schemes, achieving improvements of 51% to 87% over the baseline without MP, albeit at the cost of a 3.5% increase in BD-rate.

## 4.5 COMPARISON WITH STATE-OF-THE-ART NEURAL VIDEO CODECS

This section presents a comparison of the rate-distortion performance between state-of-the-art NVCs and MCR-Hybrid (B=5) in the presence of quantization. Following a quantization scheme (Koyuncu et al., 2024) designed for JPEG-AI, we quantize the decoder with W8A16. Additionally, the mixed-precision quantization introduced in the previous section is applied to MCR-Hybrid (B=5) to validate the effectiveness. The encoder-only components remain at 32-bit floating point. Table 7 summa-

Table 7: The complexity and coding performance comparison with the SOTA learned video codecs. The anchor is VTM 17.0 (Low-delay B). Refer to Section 4.4 for details on the complexity metrics.

| Methods | Decoder BO (G/pixel) | WPM (Channels) | Buffer size (Channels) | Model Size (MB) | BD-Rate |
|---|---|---|---|---|---|
| DCVC-FM (FP32) | 0.89 | 192 | 51.75 | 69.95 | -19.1 |
| DCVC-FM (Dec. W8A16) | 0.14 | 96 | 25.875 | 24.17 | -10.2 |
| DCVC-RT (FP16) | 0.04 | 20 | 2 | 39.47 | 5.0 |
| DCVC-RT (Dec. W8A16) | 0.02 | 20 | 2 | 22.38 | 25.9 |
| MCR-Hybrid B=5 (FP32) | 0.89 | 192 | 7.875 | 63.20 | -26.4 |
| MCR-Hybrid B=5 (Dec. W8A16) | 0.12 | 96 | 3.938 | 27.55 | -18.0 |
| MCR-Hybrid B=5 (Dec. MP, PTQ) | 0.12 | 84 | 2.476 | 29.78 | -21.7 |
| MCR-Hybrid B=5 (Dec. MP, QAT) | 0.12 | 84 | 2.476 | 29.78 | -23.7 |

Table 8: BD-rate (%) comparison for UVG and HEVC-B. All frames are tested under intra-period -1 using BT.601 for the YUV-to-RGB color conversion.

| Methods | UVG | HEVC-B | Average |
|---|---|---|---|
| DCVC-FM (FP32) | 0.0 | 0.0 | 0.0 |
| DCVC-FM (Dec. W8A16) | 20.3 | 11.1 | 15.7 |
| MCR-Hybrid B=5 (FP32) | -2.5 | -10.2 | -6.35 |
| MCR-Hybrid B=5 (Dec. W8A16) | 7.2 | -0.4 | 3.4 |
| MCR-Hybrid B=5 (Dec. MP, PTQ) | 10.2 | -3.9 | 3.15 |
| MCR-Hybrid B=5 (Dec. MP, QAT) | 0.9 | -7 | -3.05 |

rizes the complexity characteristics of these codecs. Notably, the mixed quantization approach described in Section 4.4 outperforms W8A16 at a similar complexity level for MCR-Hybrid. With the mixed quantization, MCR-Hybrid (B=5, Dec. MP, PTQ) achieves comparable coding performance to DCVC-FM (FP32), yet requires significantly lower complexity across multiple metrics. QAT further improves the performance of MP schemes. The low-complexity codec DCVC-RT achieves the minimal decoder BO, weighted peak memory, and buffer size; however, it incurs a substantial performance drop with W8A16 compared with MCR-Hybrid B=5 (Dec. W8A16) under the same settings.

### 4.6 ACCUMULATED QUANTIZATION ERRORS

In this section, we further investigate the accumulated quantization errors in terms of BD-rate increases over full sequences (intra-period = $-1$). Table 8 exhibits trends that are largely consistent with those observed for the 96-frame sequences (see Table 7). Below, we summarize the common temporal trends of quantization-induced errors for both full and 96-frame sequences: (1) Quantized codecs consistently underperform their FP32 counterparts, with performance degradation becoming more pronounced on full sequences; (2) Mixed-precision quantization (Dec. MP, QAT) achieves better coding performance than fixed-precision W8A16 quantization while preserving comparable complexity; (3) In terms of coding performance (BD-rate savings), the models follow the order MCR-Hybrid B=5 (FP32) > MCR-Hybrid B=5 (Dec. MP, QAT) > DCVC-FM (FP32) for both full and 96-frame sequences. We also illustrate the temporal degradation in PSNR caused by quantization in the Appendix (Section 6.17). It is seen that mixed-precision quantization with quantization-aware training is more effective than fixed-precision quantization.

## 5 CONCLUSION

This work presents the first systematic study of quantization effects across multiple neural video coding frameworks and temporal buffering strategies. We contrast these coding frameworks and conceptualize the temporal buffering strategies as distinct ways of creating an RNN along the temporal dimension for video coding. Extensive analyzes are conducted to understand how these coding frameworks, when combined with various temporal buffering strategies, respond to quantization. Our major findings are as follows: (1) the inter-frame main decoder exhibits the highest sensitivity to quantization, (2) the hybrid buffering strategy is superior to the implicit and explicit variants in terms of coding efficiency and buffer usage, (3) CC requires a higher-precision inter-frame decoder than MCR, and (4) mixed-precision quantization is able to strike a better balance between coding performance and complexity than fixed-precision quantization. These conclusions hold true for both PTQ and QAT.

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

## 6 APPENDIX

This supplementary document provides the following additional materials and results to assist with the understanding of our work: On the Quantization of Neural Video Codecs.

- Notations for different coding variants in Section 6.1;
- BD-rate comparisons among coding variants in Section 6.2;
- Suggested precision in Section 6.3;
- Complexity metrics in Section 6.4;
- Complexity analysis for MCR-Hybrid and CC-Hybrid in Section 6.5;
- More rate-distortion comparisons with state-of-the-art methods in Section 6.6;
- Additional results on the sensitivity of decoder components are reported in 6.7;
- Different quantization schemes in Section 6.8;
- Configurations of VTM 17.0 in Section 6.9;
- Training details in Section 6.10;
- Detailed architectures in Section 6.11;
- Visual quality comparison in Section 6.12;
- Motion error visualization in Section 6.13;
- Additional weights and activations analysis in Section 6.14;
- BD-rate increase vs. model size in Section 6.15;
- Quantizing on Transformer-based codec in Section 6.16;
- Full sequence testing in Section 6.17;
- License of Assets in Section 6.18;
- Use of Large Language Models in Section 6.19;

### 6.1 NOTATIONS FOR DIFFERENT CODING VARIANTS

Table 9 summarizes the notation used for these variants.

### 6.2 COMPARISONS OF BD-RATE AMONG CODING VARIANTS

Table 10 presents the coding performance of different variants mentioned in the main paper.

### 6.3 SUGGESTED PRECISION

As an example, to select the minimum bit-width for the inter-frame main decoder, we identify the activation and weight configurations that result in a BD-rate increase of less than 1.0%. For instance, Table 2a shows that 14-bit activation quantization (i.e. W16A14) satisfies this criterion. Similarly, Table 2b suggests that 10-bit weight quantization (i.e. W10A16) is a suitable choice. Based on these findings, we adopt W10A14 quantization for the inter-frame main decoder in Table 11.

### 6.4 COMPLEXITY METRICS

As discussed in Section 4.4 in the main paper, the complexity of the quantized codec is measured in various metrics including Decoder bit operations (BO) (Mart van Baalen, 2020) per pixel, Weighted Peak Memory, Buffer size, and Model size.

**Decoder BO per pixel:** the total number of bit operations is calculated similarly to (Mart van Baalen, 2020), as such: $BO = \sum_{i=1}^{L} bits(l_i^w) bits(l_i^a) MAC(l_i)$ where $bits(l_i^w)$, $bits(l_i^a)$ represent the bit-widths of the weights and activations for layer $l_i$.

**Weighted Peak Memory:** The Weighted Peak Memory (WPM) is computed by measuring the largest feature size in terms of the number of 32-bit full-resolution feature maps. For example,

without quantization, the largest feature size in CC-Hybrid belongs to the inter-frame main decoder occupies 192 32-bit full-resolution feature maps. When quantizing the activations of the inter-frame main decoder to INT14, the corresponding WPM is 84 (channels).

**Buffer size:** Buffer Size is expressed as the number of 32-bit full-resolution feature maps stored in the buffer. For example, the FP32 MCR-Hybrid (B=5) variant buffers 2.375 channels for the motion codec, 5 channels for the temporal buffer, and 0.5 channels for auxiliary information in the hyperprior component, taking 7.875 channels in total.

**Model size:** The model size is calculated as $\sum_{i=1}^{L} bytes(l_i^w) Params(l_i)$.

## 6.5 COMPLEXITY ANALYSIS FOR MCR-HYBRID AND CC-HYBRID

The complexity savings for MCR-Hybrid and CC-Hybrid mixed-precision configurations are presented in Table 12. Further details on the complexity metrics can be found in Section 6.4.

## 6.6 MORE STATE-OF-THE-ART RESULTS

In this section, we provide additional rate-distortion performance comparisons with state-of-the-art neural video codecs for BT.601 (see Fig. 6) and BT.709 (see Fig. 7) color space conversions. Detailed BD-rate savings are shown in Tables 13 and 14. The trade-offs between complexity and BD-rate savings for these methods in BT.601 and BT.709 color spaces are reported in Figs. 8 and 9. Additionally, the rate-distortion performance with intra-period -1 can be found in Fig. 10, and the corresponding BD-rate savings are detailed in Table 15.

## 6.7 ADDITIONAL ANALYSIS ON THE SENSITIVITY OF DECODER COMPONENTS

Additional results in Section and 4.3 for other variants (MCR-Implicit, MCR-Explicit, CC-Implicit, CC-Explicit) are reported in Fig. 11, Fig. 12 and Table 16, 17.

## 6.8 DIFFERENT QUANTIZATION SCHEME

Table 18 provides different quantization schemes such as MSE and MinMax quantization.

## 6.9 VTM 17.0 CONFIGURATIONS VTM

Following the recommendation from Li et al. (2023), we encode videos in YUV444 format. We use the *encoder_lowdelay_vtm.cfg* of VTM (vtm) with the following parameters:

–c {config file name}

–InputFile={input file name}

–InputBitDepth=8

–InputChromaFormat=444

–ChromaFormatIDC=444

–InternalBitDepth=10

–OutputBitDepth=8

–DecodingRefreshType=2

–FrameRate={frame rate}

–FrameSkip=0

–SourceWidth={width}

–SourceHeight={height}

–FramesToBeEncoded=96

–Level=4.1

–IntraPeriod=32

–QP={qp}

–BitstreamFile={bitstream file name}

–ReconFile={reconstruction file name}

## 6.10 TRAINING DETAILS

Tables 19, 20, and 21 present the training recipes for the coding variants discussed in the main paper, following the setup described in Chen et al. (2024b). Variants that share the same buffering strategy are trained using an identical procedure. To ensure a fair comparison, the final training stage for all buffering strategies is extended until convergence. All training experiments are conducted on an Intel(R) Xeon(R) Platinum 8480C processor with either an NVIDIA H200 or NVIDIA H100 GPU.

## 6.11 NETWORK ARCHITECTURE DETAILS

Fig. 13, 14, 15, 16, and 17 provide additional details on MCR-Hybrid and CC-Hybrid. Other coding variants share similar components with MCR-Hybrid, and CC-Hybrid. The Entropy Model in Fig.13, the Condition Network in Fig.15, and the Mask Generator in Fig.17 adopt the architecture proposed in Chen et al. (2024b). The Refinement Network in Fig.13 follows the Frame Generator design from Li et al. (2024).

## 6.12 VISUAL QUALITY COMPARISON

We provide some visualization results for MCR/CC-Hybrid and DCVC-FM in Fig.18.

## 6.13 VISUALIZATION FOR MOTION ERROR

Fig. 19 shows that the quantized motion codec introduces larger motion error than the floating-point motion codec.

## 6.14 ADDITIONAL WEIGHTS AND ACTIVATIONS ANALYSIS

We have visualized the histograms of the weights and activations of each decoder component of MCR-Hybrid under W8A10 quantization in Fig 20, 21.

## 6.15 BD-RATE INCREASE VS. MODEL SIZE

In Fig. 22, we provide a comparison between each component's model size and the corresponding BD-rate increase under 8-bit weight and 10-bit activation quantization.

## 6.16 QUANTIZING ON TRANSFORMER-BASED CODEC

We provide a component-wise quantization sensitivity analysis for the Transformer-based codec MaskCRT (Chen et al., 2024b). MaskCRT follows the masked conditional residual coding scheme with an explicit buffer (MCR-Explicit). In Fig. 23, similar to the MCR-Explicit with DCVC-FM backbones, MaskCRT exhibits a similar trend in BD-rate increases, reinforcing our claims regarding the relative component-wise sensitivity to quantization in our work.

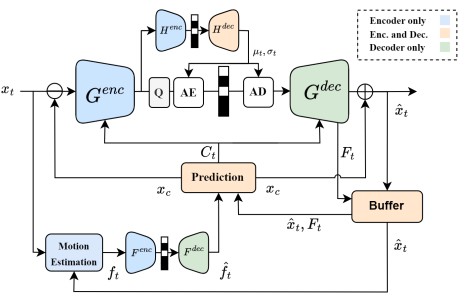

(a) Masked Condtional Residual Coding with Hybrid buffer

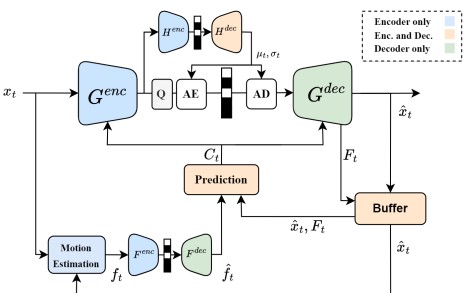

(b) Condtional Coding with Hybrid buffer

Figure 5: High-level architecture of the coding frameworks. For brevity, entropy coding for hyperprior and motion is omitted. The colors of the components in the figure indicate whether the component is included only in the encoder, only in the decoder, or in both.

Table 9: Notation of coding variants used in the experiments.

| Variants | Coding Schemes | | Buffer | | |
|---|---|---|---|---|---|
| | MCR | CC | Explicit | Implicit | Hybrid |
| MCR-Explicit | ✓ | | ✓ | | |
| MCR-Implicit | ✓ | | | ✓ | |
| MCR-Hybrid | ✓ | | | | ✓ |
| CC-Explicit | | ✓ | ✓ | | |
| CC-Implicit | | ✓ | | ✓ | |
| CC-Hybrid | | ✓ | | | ✓ |

In Fig. 24, we also present a comparison between each component's model size and its corresponding BD-rate increase under 8-bit weight and 10-bit activation quantization for both MaskCRT and our masked conditional residual coding framework with explicit buffers (MCR-Explicit).

### 6.17 Full sequence testing

We present the temporal degradation in Fig. 25 to illustrate how quantization affects frame quality over time.

### 6.18 License of assets used

Table 22 summarizes the used assets in our work along with their license terms.

### 6.19 Use of large language models

As the authors are not native speakers, we composed the sentences ourselves and use ChatGPT to check the grammar and polish them.

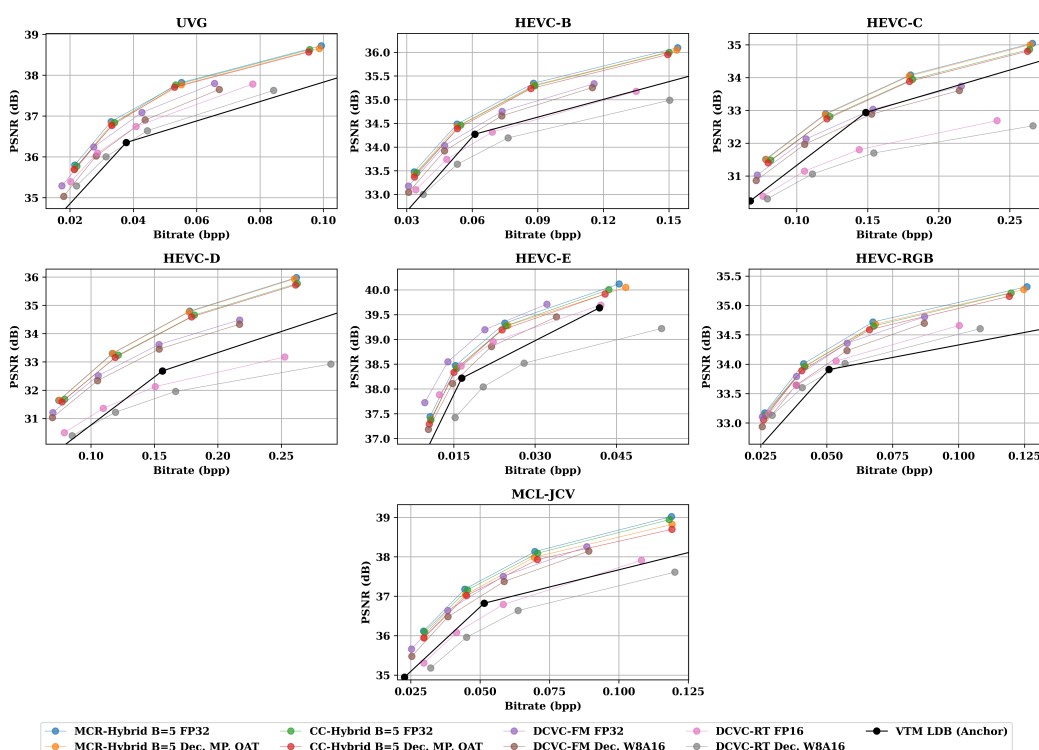

Figure 6: Rate-distortion performance comparison with state-of-the-arts video codecs under intra-period 32 using BT.601 as colorspace.

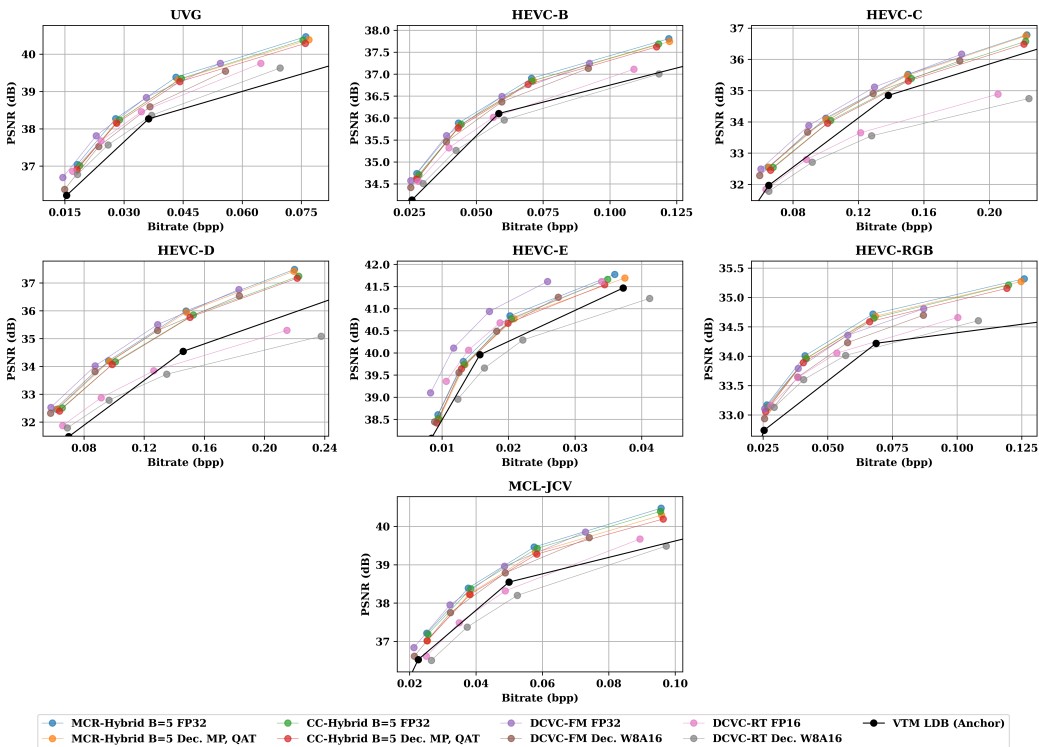

Figure 7: Rate-distortion performance comparison with state-of-the-arts video codecs under intra-period 32 using BT.709 as colorspace.

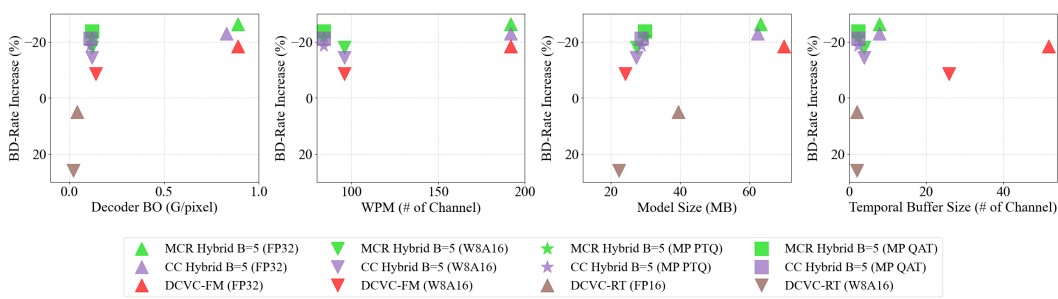

Figure 8: Complexity-performance trade-offs in BT.601 colorspace. The complexity metrics include the decoder BO, weighted peak memory, model size, and temporal buffer size. The vertical axis is the BD-rate savings in terms of PSNR-RGB evaluated on all testing datasets with VTM 17.0 as an anchor.

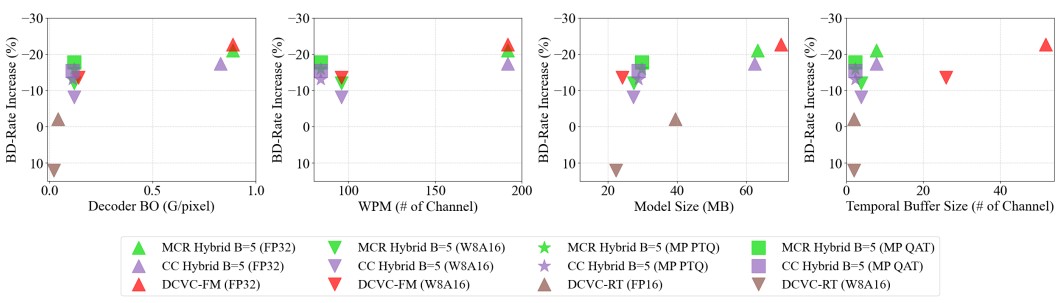

Figure 9: Complexity-performance trade-offs in BT.709 colorspace. The complexity metrics include the decoder BO, weighted peak memory, model size, and temporal buffer size. The vertical axis is the BD-rate savings in terms of PSNR-RGB evaluated on all testing datasets with VTM 17.0 as an anchor.

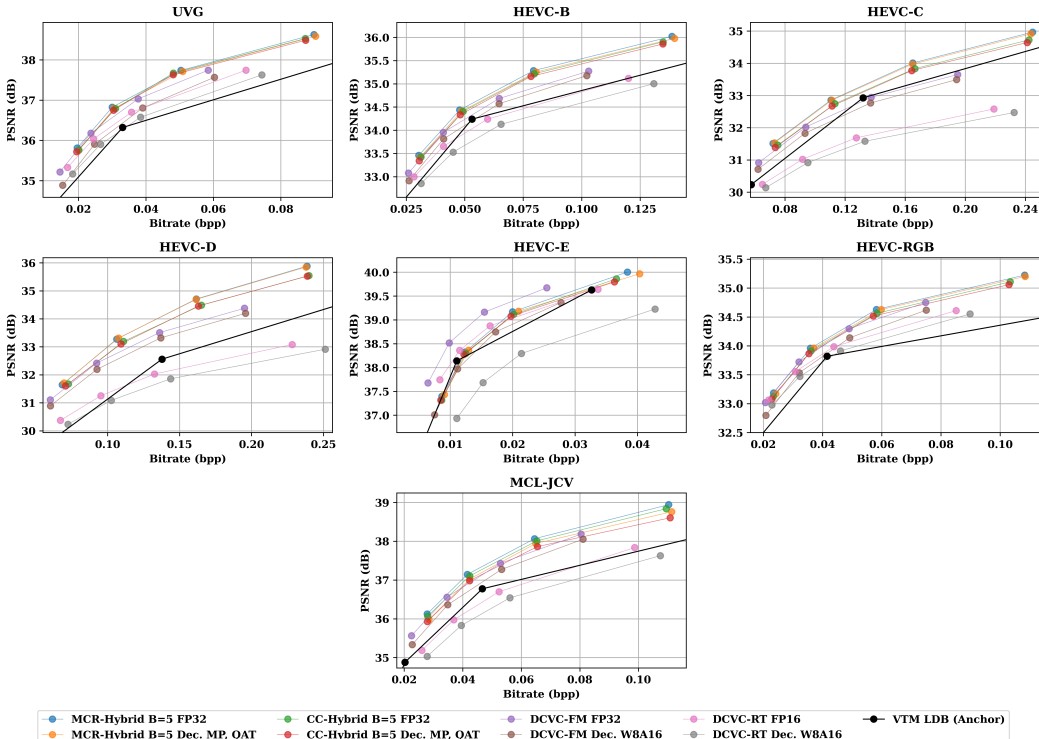

Figure 10: Rate-distortion performance comparison with state-of-the-arts video codecs under intra-period -1 using BT.601 as colorspace.

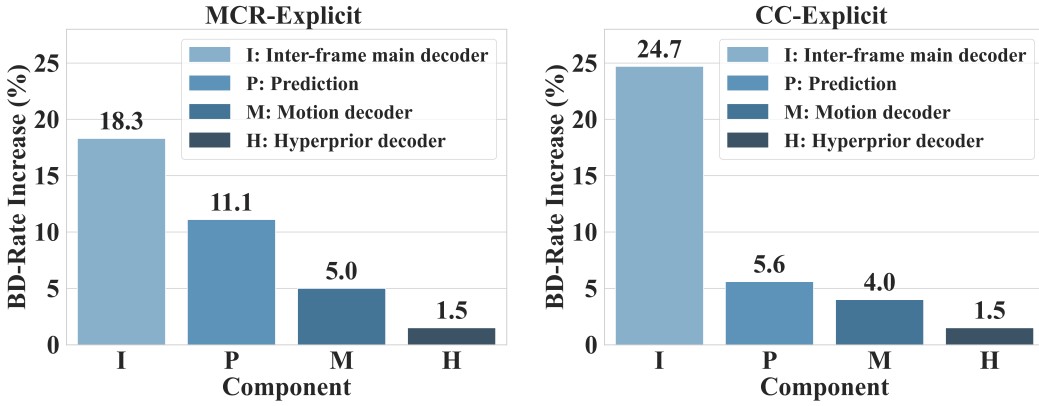

Figure 11: Visualization of the BD-rate increases due to quantization. The component-wise analysis of quantizing MCR-Explicit and CC-Explicit to W8A10. The BD-rate increases are measured with the FP32 counterparts serving as anchors.

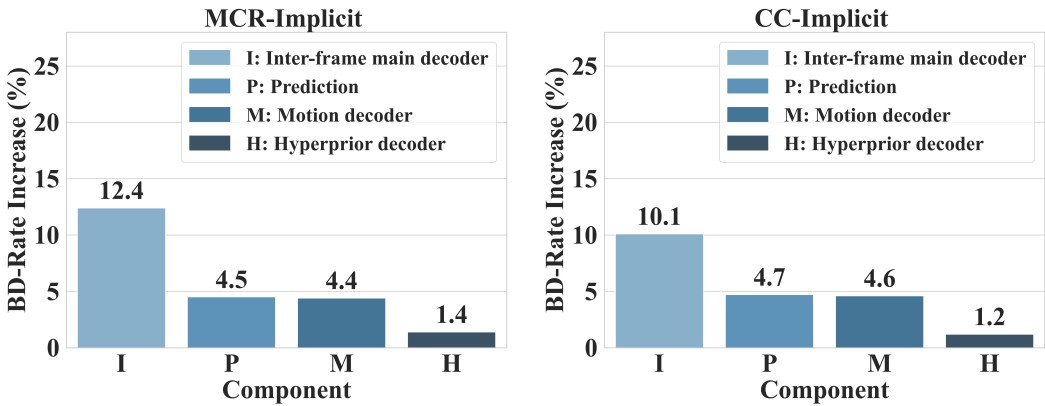

Figure 12: Visualization of the BD-rate increases due to quantization. The component-wise analysis of quantizing MCR-Implicit and CC-Implicit to W8A10. The BD-rate increases are measured with the FP32 counterparts serving as anchors.

Table 10: BD-rate (%) comparison of different coding variants (FP32) with the temporal buffer B=5 in terms of PSNR-RGB using BT.601 as colorspace. The anchor is MCR-Hybrid.

|  | UVG | HEVC-B | HEVC-C | HEVC-D | HEVC-E | HEVC-RGB | MCL-JCV | Avg. |
|---|---|---|---|---|---|---|---|---|
| MCR-Hybrid | 0.0 | 0.0 | 0.0 | 0.0 | 0.0 | 0.0 | 0.0 | 0.0 |
| MCR-Implicit | 10.6 | 6.6 | 5.8 | 5.7 | 16.6 | 8.7 | 4.6 | 8.4 |
| MCR-Explicit | 4.7 | 8.4 | 16.1 | 14.6 | 11.5 | 4.8 | 5.6 | 9.4 |
| CC-Hybrid | 0.0 | 0.0 | 0.0 | 0.0 | 0.0 | 0.0 | 0.0 | 0.0 |
| CC-Implicit | 8.1 | 6.6 | 11.1 | 7.8 | 11.6 | 5.2 | 4.8 | 7.9 |
| CC-Explicit | 14.4 | 18.2 | 26.8 | 23.6 | 22.1 | 15.9 | 12.8 | 19.1 |

Table 11: Mixed-precision quantization for MCR-Hybrid and CC-Hybrid (B=5) decoders.

| Model | Components | Inter-frame main decoder | Prediction | Motion decoder | Hyperprior decoder | Buffer |
|---|---|---|---|---|---|---|
| MCR-Hybrid | W-bits | 10 | 10 | 10 | 8 | - |
|  | A-bits | 14 | 12 | 14 | 12 | 8 |
| CC-Hybrid | W-bits | 10 | 10 | 8 | 8 | - |
|  | A-bits | 14 | 12 | 12 | 12 | 8 |

Table 12: The complexity and coding performance analysis for MCR-Hybrid B=5 and CC-Hybrid B=5 mixed-precision quantization. "w/ MP" denotes mixed-precision quantization. "w/o MP" is the FP32 model. WPM: Weighted Peak Memory. Refer to Section 6.4 for details on the complexity metrics.

| Method | Setting | Decoder BO (G/pixel) | WPM (Channels) | Buffer size (Channels) | Model Size (MB) | BD-Rate |
|---|---|---|---|---|---|---|
| MCR-Hybrid | w/o MP | 0.89 | 192 | 7.875 | 63.20 | 0.0 (Anchor) |
| | w/ MP | 0.12 | 84 | 2.476 | 29.78 | 3.5 |
| | Savings | 87% | 51% | 69% | 53% | - |
| CC-Hybrid | w/o MP | 0.83 | 192 | 7.875 | 62.4 | 0.0 (Anchor) |
| | w/ MP | 0.11 | 84 | 2.328 | 28.8 | 2.3 |
| | Savings | 87% | 51% | 70% | 54% | - |

Table 13: BD-rate (%) comparison with state-of-the-arts video codecs under intra-period 32 using BT.601 as colorspace.

| | UVG | HEVC-B | HEVC-C | HEVC-D | HEVC-E | HEVC-RGB | MCL-JCV | Avg. |
|---|---|---|---|---|---|---|---|---|
| VTM (LDB) | 0.0 | 0.0 | 0.0 | 0.0 | 0.0 | 0.0 | 0.0 | 0.0 |
| DCVC-FM (FP32) | -23.7 | -10.2 | -5.3 | -26.9 | -29.4 | -18.4 | -19.7 | -19.1 |
| DCVC-FM (Dec. W8A16) | -12.7 | -5.1 | -1.5 | -23.6 | -6.4 | -9.8 | -13.1 | -10.3 |
| DCVC-RT (FP16) | -12.6 | 9.5 | 31.3 | 9.0 | -11.3 | -5.4 | 14.8 | 5.0 |
| DCVC-RT (Dec. W8A16) | 1.0 | 28.9 | 43.8 | 25.2 | 43.1 | 3.5 | 35.7 | 25.9 |
| MCR-Hybrid B=5 (FP32) | -33.9 | -22.6 | -16.2 | -35.5 | -19.7 | -26.9 | -30.0 | -26.4 |
| MCR-Hybrid B=5 (Dec. W8A16) | -26.9 | -17.6 | -11.6 | -31.4 | 2.9 | -20.9 | -20.7 | -18.0 |
| MCR-Hybrid B=5 (Dec. MP, PTQ) | -29.2 | -19.7 | -14.5 | -34.1 | -11.3 | -22.6 | -20.4 | -21.7 |
| MCR-Hybrid B=5 (Dec. MP, QAT) | -31.4 | -20.2 | -15.6 | -35.2 | -16.1 | -23.5 | -24.0 | -23.7 |
| CC-Hybrid B=5 (FP32) | -31.9 | -19.2 | -11.9 | -31.9 | -16.2 | -22.9 | -27.6 | -23.1 |
| CC-Hybrid B=5 (Dec. W8A16) | -22.8 | -14.4 | -7.8 | -27.8 | 4.7 | -15.6 | -16.2 | -14.3 |
| CC-Hybrid B=5 (Dec. MP, PTQ) | -28.9 | -16.8 | -10.2 | -30.4 | -9.9 | -18.9 | -15.4 | -18.6 |
| CC-Hybrid B=5 (Dec. MP, QAT) | -30.8 | -17.9 | -11.5 | -31.6 | -15.1 | -20.5 | -21.3 | -21.2 |

Table 14: BD-rate (%) comparison with state-of-the-arts video codecs under intra-period 32 using BT.709 as colorspace.

| | UVG | HEVC-B | HEVC-C | HEVC-D | HEVC-E | HEVC-RGB | MCL-JCV | Avg. |
|---|---|---|---|---|---|---|---|---|
| VTM (LDB) | 0.0 | 0.0 | 0.0 | 0.0 | 0.0 | 0.0 | 0.0 | 0.0 |
| DCVC-FM (FP32) | -24.0 | -16.0 | -14.8 | -31.0 | -31.8 | -22.6 | -18.5 | -22.7 |
| DCVC-FM (Dec. W8A16) | -11.9 | -10.7 | -10.6 | -27.6 | -8.4 | -14.5 | -11.1 | -13.5 |
| DCVC-RT (FP16) | -13.6 | 0.0 | 17.6 | 0.5 | -16.3 | -10.5 | 7.3 | -2.1 |
| DCVC-RT (Dec. W8A16) | -2.6 | 10.4 | 26.9 | 10.3 | 20.3 | -2.1 | 21.7 | 12.1 |
| MCR-Hybrid B=5 (FP32) | -25.6 | -18.0 | -10.9 | -28.0 | -12.9 | -29.7 | -21.7 | -21.0 |
| MCR-Hybrid B=5 (Dec. W8A16) | -17.6 | -12.3 | -5.9 | -23.2 | 12.0 | -24.0 | -12.9 | -12.0 |
| MCR-Hybrid B=5 (Dec. MP, PTQ) | -19.5 | -14.7 | -8.9 | -26.2 | -3.4 | -25.6 | -12.8 | -15.9 |
| MCR-Hybrid B=5 (Dec. MP, QAT) | -21.8 | -14.9 | -10.1 | -27.2 | -8.0 | -26.5 | -15.1 | -17.7 |
| CC-Hybrid B=5 (FP32) | -22.2 | -15.0 | -6.5 | -23.7 | -9.4 | -26.1 | -19.2 | -17.4 |
| CC-Hybrid B=5 (Dec. W8A16) | -11.4 | -9.5 | -1.8 | -18.8 | 13.4 | -19.2 | -9.3 | -8.1 |
| CC-Hybrid B=5 (Dec. MP, PTQ) | -18.5 | -12.2 | -4.4 | -22.0 | -2.2 | -22.3 | -9.1 | -13.0 |
| CC-Hybrid B=5 (Dec. MP, QAT) | -20.1 | -13.6 | -5.7 | -23.4 | -7.8 | -23.9 | -13.3 | -15.4 |

Table 15: BD-rate (%) comparison with state-of-the-arts video codecs under intra-period -1 using BT.601 as colorspace.

| | UVG | HEVC-B | HEVC-C | HEVC-D | HEVC-E | HEVC-RGB | MCL-JCV | Avg. |
|---|---|---|---|---|---|---|---|---|
| VTM (LDB) | 0.0 | 0.0 | 0.0 | 0.0 | 0.0 | 0.0 | 0.0 | 0.0 |
| DCVC-FM (FP32) | -23.5 | -8.4 | -2.0 | -27.7 | -28.5 | -18.3 | -19.1 | -18.2 |
| DCVC-FM (Dec. W8A16) | -8.4 | -1.6 | 3.4 | -23.1 | 8.8 | -6.2 | -10.6 | -5.4 |
| DCVC-RT (FP16) | -13.0 | 11.0 | 37.4 | 8.5 | -6.5 | -7.4 | 15.9 | 6.6 |
| DCVC-RT (Dec. W8A16) | 0.8 | 29.8 | 48.2 | 23.4 | 75.7 | 3.1 | 34.6 | 30.8 |
| MCR-Hybrid B=5 (FP32) | -31.6 | -19.7 | -12.7 | -35.5 | -2.2 | -25.2 | -29.2 | -22.3 |
| MCR-Hybrid B=5 (Dec. W8A16) | -22.6 | -12.5 | -6.7 | -30.2 | 35.8 | -16.0 | -18.6 | -10.1 |
| MCR-Hybrid B=5 (Dec. MP, PTQ) | -25.1 | -15.8 | -10.5 | -33.7 | 13.7 | -20.1 | -18.2 | -15.7 |
| MCR-Hybrid B=5 (Dec. MP, QAT) | -29.1 | -16.9 | -11.8 | -35.3 | 0.9 | -22.8 | -22.8 | -19.7 |
| CC-Hybrid B=5 (FP32) | -29.4 | -15.6 | -7.1 | -30.7 | 1.4 | -20.8 | -26.1 | -18.3 |
| CC-Hybrid B=5 (Dec. W8A16) | -18.0 | -9.5 | -0.3 | -26.2 | 33.6 | -11.0 | -13.1 | -6.4 |
| CC-Hybrid B=5 (Dec. MP, PTQ) | -26.1 | -12.3 | -4.7 | -28.2 | 11.6 | -16.5 | -12.9 | -12.7 |
| CC-Hybrid B=5 (Dec. MP, QAT) | -28.7 | -14.0 | -6.5 | -30.6 | 1.0 | -19.1 | -19.5 | -16.8 |

Table 16: Effects of quantizing each decoder component in MCR-Explicit and CC-Explicit, with FP32 as the respective anchor.

(a) Effect of quantizing **activations (A)** for each decoder component.

| Component | MCR-Explicit | | | | | CC-Explicit | | | | |
|---|---|---|---|---|---|---|---|---|---|---|
| | FP32 | W16A14 | W16A12 | W16A10 | W16A8 | FP32 | W16A14 | W16A12 | W16A10 | W16A8 |
| **Inter-frame main decoder** | | 0.3 | 2.4 | 14.4 | – | | 0.4 | 0.9 | 11.7 | – |
| **Prediction** | 0.0 | 0.0 | 0.1 | 5.0 | – | 0.0 | 0.0 | 0 | 0.6 | – |
| **Motion decoder** | (Anchor) | 1.2 | 1.4 | 5.0 | – | (Anchor) | 0.1 | 0.4 | 4.3 | – |
| **Hyperprior decoder** | | 0.0 | 0.2 | 1.4 | – | | 0.0 | 0.3 | 1.2 | – |

(b) Effect of quantizing **network weights (W)** for each decoder component.

| Component | MCR-Explicit | | | | | CC-Explicit | | | | |
|---|---|---|---|---|---|---|---|---|---|---|
| | FP32 | W14A16 | W12A16 | W10A16 | W8A16 | FP32 | W14A16 | W12A16 | W10A16 | W8A16 |
| **Inter-frame main decoder** | | – | 0.2 | 1.1 | 4.5 | | – | 0.2 | 0.1 | 14.9 |
| **Prediction** | 0.0 | – | 0.2 | 1.2 | 8.9 | 0.0 | – | 0.1 | 0.4 | 5.3 |
| **Motion decoder** | (Anchor) | – | 1.1 | 1.1 | 1.2 | (Anchor) | – | 0.1 | 0.2 | 0.2 |
| **Hyperprior decoder** | | – | 0.0 | 0.0 | 0.0 | | – | 0.0 | 0.0 | 0.0 |

Table 17: Effects of quantizing each decoder component in MCR-Implicit and CC-Implicit, with FP32 as the respective anchor.

(a) Effect of quantizing **activations (A)** for each decoder component.

| Component | MCR-Implicit | | | | | CC-Implicit | | | | |
|---|---|---|---|---|---|---|---|---|---|---|
| | FP32 | W16A14 | W16A12 | W16A10 | W16A8 | FP32 | W16A14 | W16A12 | W16A10 | W16A8 |
| **Inter-frame main decoder** | | 0.3 | 1.4 | 12.4 | – | | 0.5 | 1.1 | 9.6 | – |
| **Prediction** | 0.0 | 0.0 | 0.1 | 1.4 | – | 0.0 | 0.0 | 0.1 | 0.9 | – |
| **Motion decoder** | (Anchor) | 0.6 | 0.8 | 4.2 | – | (Anchor) | 0.5 | 1.2 | 4.7 | – |
| **Hyperprior decoder** | | 0.0 | 0.2 | 1.5 | – | | 0.0 | 0.2 | 1.2 | – |

(b) Effect of quantizing **network weights (W)** for each decoder component.

| Component | MCR-Implicit | | | | | CC-Implicit | | | | |
|---|---|---|---|---|---|---|---|---|---|---|
| | FP32 | W14A16 | W12A16 | W10A16 | W8A16 | FP32 | W14A16 | W12A16 | W10A16 | W8A16 |
| **Inter-frame main decoder** | | – | 0.2 | 0.2 | 1.8 | | – | 0.4 | 0.5 | 1.1 |
| **Prediction** | 0.0 | – | -0.1 | 0.4 | 3.2 | 0.0 | – | 0.1 | 0.0 | 2.0 |
| **Motion decoder** | (Anchor) | – | 0.6 | 0.6 | 0.8 | (Anchor) | – | 0.4 | 0.4 | 0.7 |
| **Hyperprior decoder** | | – | 0.0 | 0.0 | 0.1 | | – | 0.0 | 0.1 | 0.2 |

Table 18: Different quantization schemes comparison. MCR-Hybrid FP32 serves as the anchor.

| Coding Schemes | MCR-Hybrid | | | |
|---|---|---|---|---|
| Component | I | P | M | H |
| MSE | 12.8 | 6.9 | 5.4 | 1.5 |
| MinMax | 47.1 | 9.6 | 6.5 | 4.8 |

Table 19: Training procedure of Explicit temporal buffer variants. MENet represents the motion estimation network. EPA is the error propagation aware training in (Lu et al., 2020). Ref represents the characteristic of reference temporal information in the inter-frame codec. TTC means "training to convergence".

| Phase | # Frames | Training Modules | Loss | lr | Epoch |
|---|---|---|---|---|---|
| Motion Coding (Ref: Explicit) | 3 | Motion codec | $R_t^{motion} + \lambda \times D(x_t, warp(x_{t-1}, \hat{f}_t))$ | 1e-4 | 8 |
| Motion Compensation (Ref: Explicit) | 3 | Prediction Network | $\lambda \times D(x_t, x_c)$ | 1e-4 | 10 |
| Inter-frame Coding (Ref: Explicit) | 2 | Inter-frame codec and mask generator | $R_t + \lambda \times D(x_t, \hat{x}_t)$ | 1e-4 | 2 |
| Motion Compensation (Ref: Explicit) | 3 | Prediction Network | $R_t + \lambda \times (D(x_t, x_c) + D(x_t, \hat{x}_t))/2$ | 1e-4 | 3 |
| Inter-frame Coding (Ref: Explicit) | 3 / 5 | All modules except MENet and Motion codec | $R_t + \lambda \times D(x_t, \hat{x}_t)$ / $R_t + \lambda \times D(x_t, \hat{x}_t)$ | 1e-4 / 1e-4 | 8 / 5 |
| Finetuning (Ref: Explicit) | 3 / 5 | All modules except MENet | $R_t + \lambda \times D(x_t, \hat{x}_t)$ / $R_t + \lambda \times D(x_t, \hat{x}_t)$ | 1e-4 / 1e-4 | 6 / 5 |
| Finetuning with EPA (Ref: Explicit) | 5 / 5 | All modules except MENet / All modules | $R_t + \lambda \times D(x_t, \hat{x}_t)$ / $R_t + \lambda \times D(x_t, \hat{x}_t)$ | 1e-5 / 1e-5 | 4 / TTC |

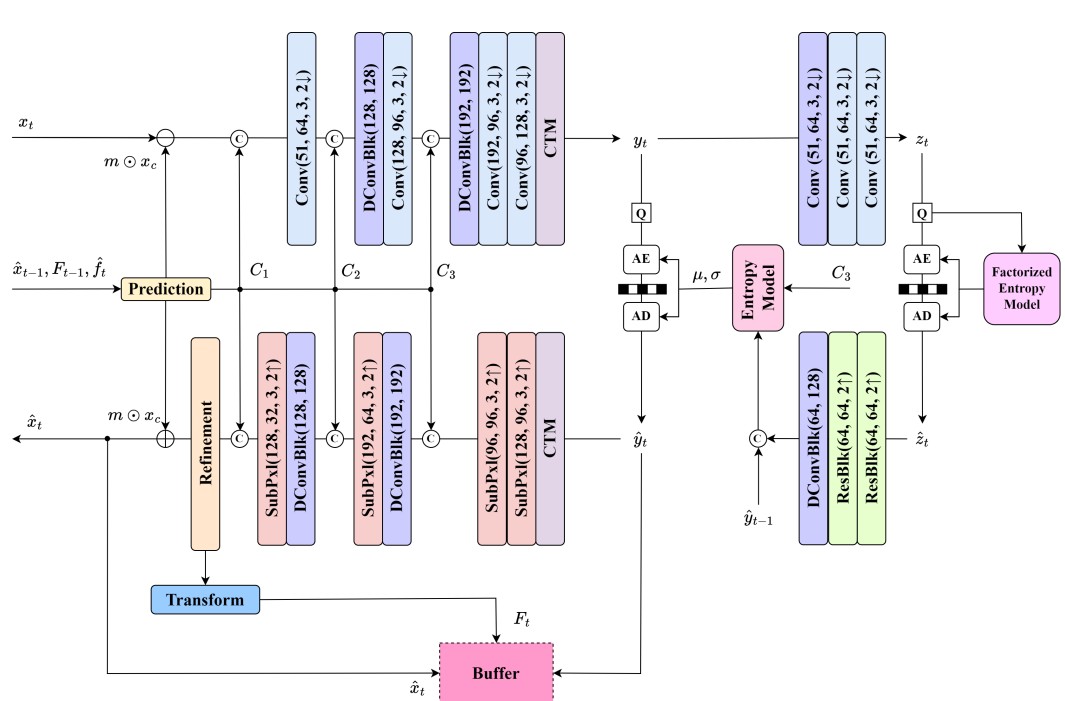

Figure 13: Network architecture detail of the inter-frame codec in MCR-Hybrid.

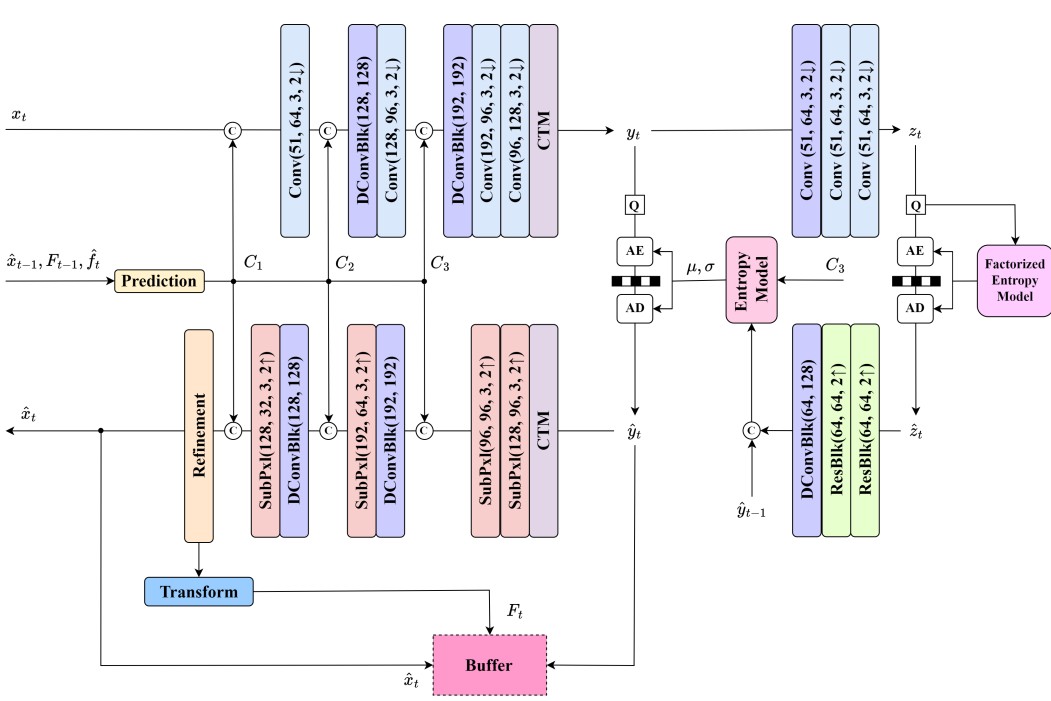

Figure 14: Network architecture detail of the inter-frame codec in CC-Hybrid.

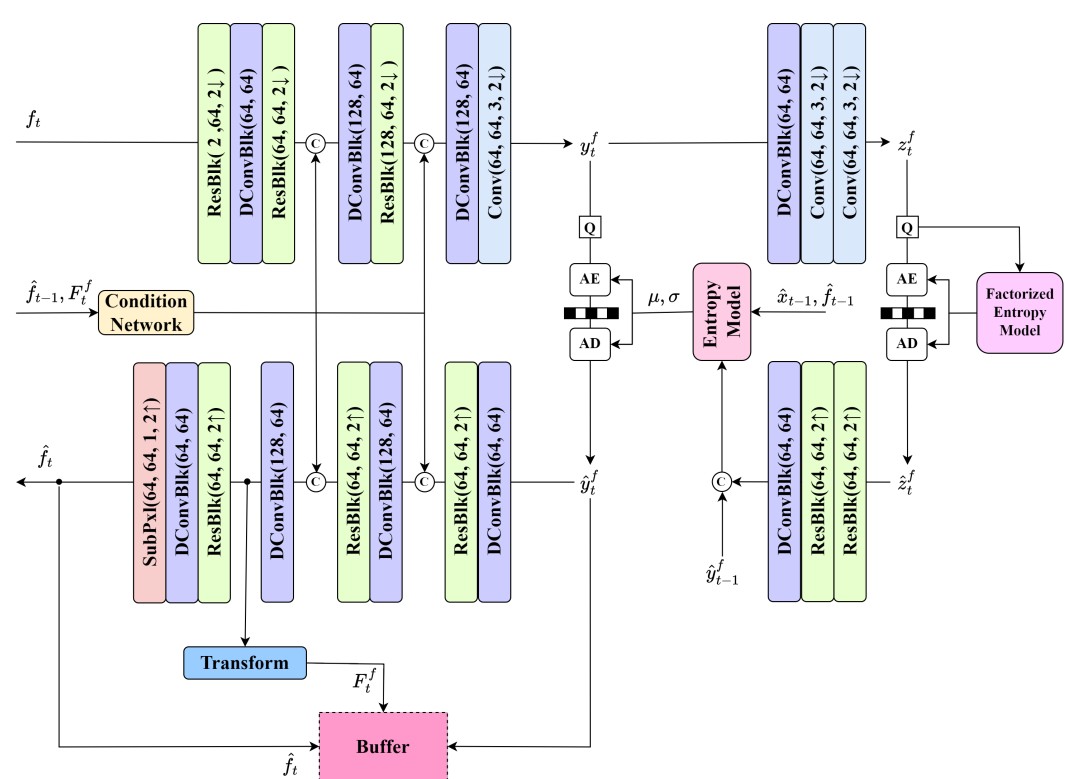

Figure 15: Network architecture detail of the motion codec.

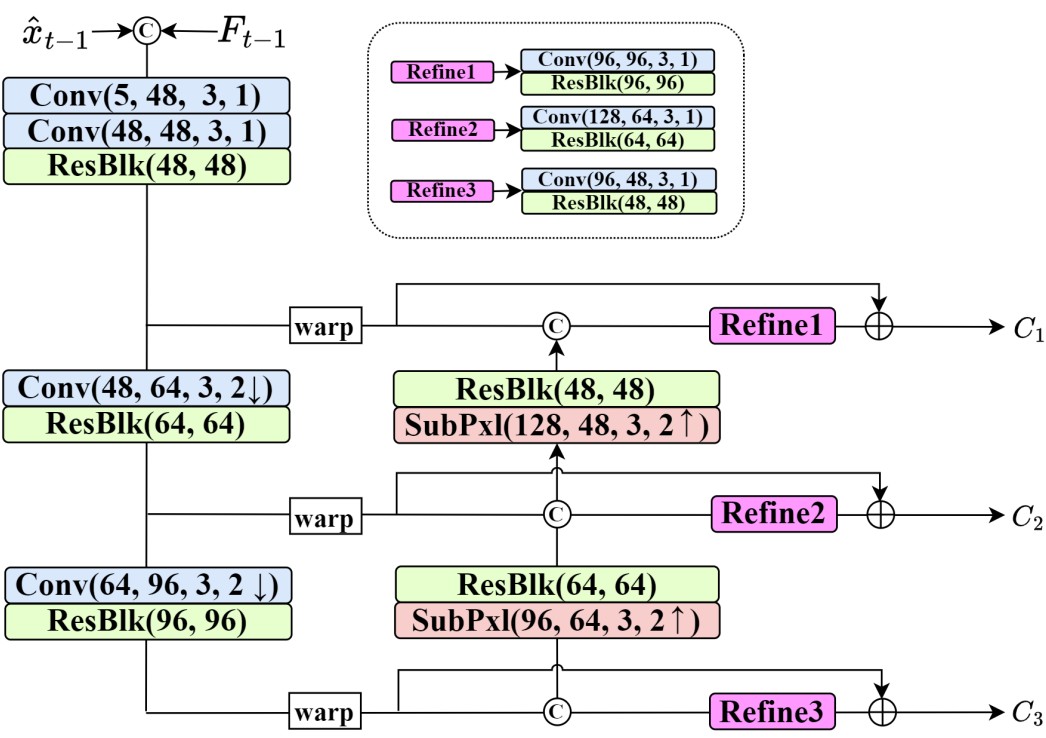

Figure 16: Network architecture detail of the prediction network in CC-Hybrid.

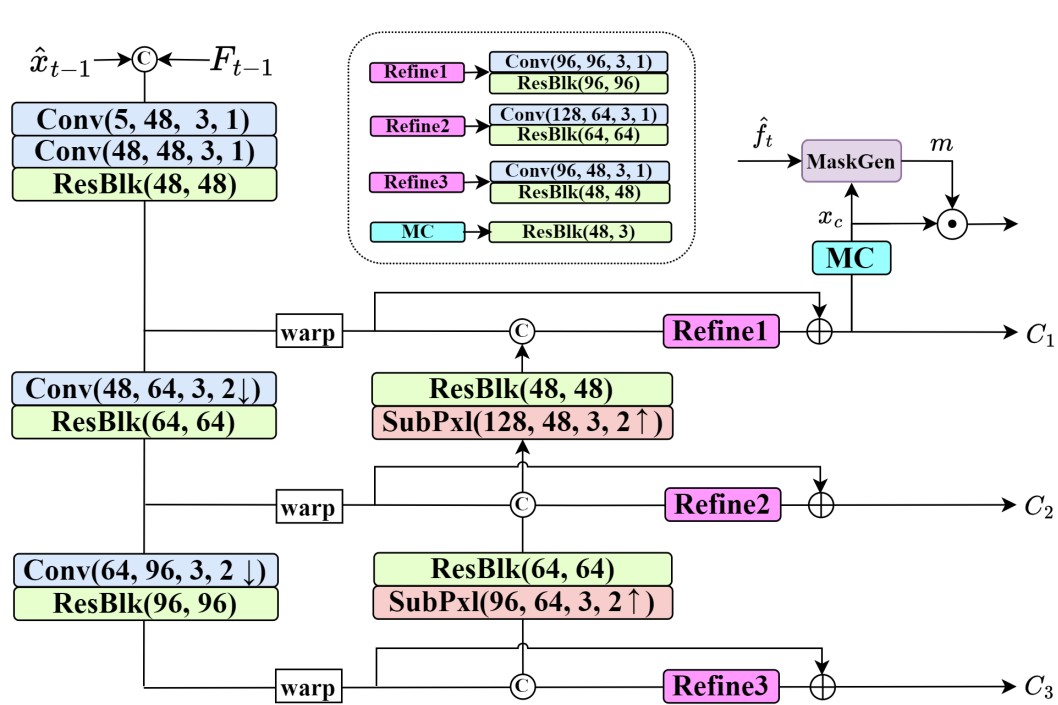

Figure 17: Network architecture detail of the prediction network in MCR-Hybrid.

Table 20: Training procedure of Implicit temporal buffer variants. MENet represents the motion estimation network. EPA is the error propagation aware training in Lu et al. (2020). Ref represents the characteristic of reference temporal buffer in the inter-frame codec. TTC means "training to convergence".

| Phase | # Frames | Training Modules | Loss | lr | Epoch |
|---|---|---|---|---|---|
| Motion Coding (Ref: Explicit) | 3 | Motion codec | $R_t^{motion} + \lambda \times D(x_t, warp(x_{t-1}, \hat{f}_t))$ | 1e-4 | 8 |
| Motion Compensation (Ref: Explicit) | 3 | Prediction Network | $\lambda \times D(x_t, x_c)$ | 1e-4 | 10 |
| Inter-frame Coding (Ref: Explicit) | 2 | Inter-frame codec and mask generator | $R_t + \lambda \times D(x_t, \hat{x}_t)$ | 1e-4 | 2 |
| Motion Compensation (Ref: Explicit) | 3 | Prediction | $R_t + \lambda \times (D(x_t, x_c) + D(x_t, \hat{x}_t))/2$ | 1e-4 | 3 |
| Inter-frame Coding (Ref: Explicit) | 3 | All modules except MENet, Motion codec and Transform | $R_t + \lambda \times D(x_t, \hat{x}_t)$ | 1e-4 | 8 |
| | 5 | | $R_t + \lambda \times D(x_t, \hat{x}_t)$ | 1e-4 | 5 |
| Finetuning (Ref: Explicit) | 3 | All modules except MENet and Transform | $R_t + \lambda \times D(x_t, \hat{x}_t)$ | 1e-4 | 6 |
| | 5 | | $R_t + \lambda \times D(x_t, \hat{x}_t)$ | 1e-4 | 5 |
| Feature Generation (Ref: Implicit) | 3 | Transform | $R_t + \lambda \times D(x_t, \hat{x}_t)$ | 1e-4 | 3 |
| Motion Compensation (Ref: Implicit) | 3 | Prediction Network | $R_t + \lambda \times (D(x_t, x_c) + D(x_t, \hat{x}_t))/2$ | 1e-4 | 4 |
| Inter-frame Coding (Ref: Implicit) | 3 | All modules except MENet and Motion codec | $R_t + \lambda \times D(x_t, \hat{x}_t)$ | 1e-4 | 8 |
| | 5 | | $R_t + \lambda \times D(x_t, \hat{x}_t)$ | 1e-4 | 4 |
| Finetuning (Ref: Implicit) | 3 | All modules except MENet | $R_t + \lambda \times D(x_t, \hat{x}_t)$ | 1e-4 | 3 |
| | 5 | | $R_t + \lambda \times D(x_t, \hat{x}_t)$ | 1e-4 | 4 |
| Finetuning with EPA (Ref: Implicit) | 5 | All modules except MENet | $R_t + \lambda \times D(x_t, \hat{x}_t)$ | 1e-5 | 4 |
| | 5 | All modules | $R_t + \lambda \times D(x_t, \hat{x}_t)$ | 1e-5 | TTC |

Table 21: Training procedure of Hybrid temporal buffer variants. MENet represents the motion estimation network. EPA is the error propagation aware training in Lu et al. (2020). Ref represents the characteristic of reference temporal buffer in the inter-frame codec. TTC means "training to convergence".

| Phase | # Frames | Training Modules | Loss | lr | Epoch |
|---|---|---|---|---|---|
| Motion Coding (Ref: Explicit) | 3 | Motion codec | $R_t^{motion} + \lambda \times D(x_t, warp(x_{t-1}, \hat{f}_t))$ | 1e-4 | 8 |
| Motion Compensation (Ref: Explicit) | 3 | Prediction Network | $\lambda \times D(x_t, x_c)$ | 1e-4 | 10 |
| Inter-frame Coding (Ref: Explicit) | 2 | Inter-frame codec and mask generator | $R_t + \lambda \times D(x_t, \hat{x}_t)$ | 1e-4 | 2 |
| Motion Compensation (Ref: Explicit) | 3 | Prediction Network | $R_t + \lambda \times (D(x_t, x_c) + D(x_t, \hat{x}_t))/2$ | 1e-4 | 3 |
| Inter-frame Coding (Ref: Explicit) | 3
5 | All modules except MENet, Motion codec and Transform in Fig. 13 | $R_t + \lambda \times D(x_t, \hat{x}_t)$
$R_t + \lambda \times D(x_t, \hat{x}_t)$ | 1e-4
1e-4 | 8
5 |
| Finetuning (Ref: Explicit) | 3
5 | All modules except MENet and Transform in Fig. 13 | $R_t + \lambda \times D(x_t, \hat{x}_t)$
$R_t + \lambda \times D(x_t, \hat{x}_t)$ | 1e-4
1e-4 | 6
5 |
| Feature Generation (Ref: Hybrid) | 3 | Transform in Fig. 13 | $R_t + \lambda \times D(x_t, \hat{x}_t)$ | 1e-4 | 3 |
| Motion Compensation (Ref: Hybrid) | 3 | Prediction Network | $R_t + \lambda \times (D(x_t, x_c) + D(x_t, \hat{x}_t))/2$ | 1e-4 | 4 |
| Inter-frame Coding (Ref: Hybrid) | 3
5 | All modules except MENet and Motion codec | $R_t + \lambda \times D(x_t, \hat{x}_t)$
$R_t + \lambda \times D(x_t, \hat{x}_t)$ | 1e-4
1e-4 | 8
4 |
| Finetuning (Ref: Hybrid) | 3
5 | All modules except MENet | $R_t + \lambda \times D(x_t, \hat{x}_t)$
$R_t + \lambda \times D(x_t, \hat{x}_t)$ | 1e-4
1e-4 | 3
4 |
| Finetuning with EPA (Ref: Hybrid) | 5
5 | All modules except MENet
All modules | $R_t + \lambda \times D(x_t, \hat{x}_t)$
$R_t + \lambda \times D(x_t, \hat{x}_t)$ | 1e-5
1e-5 | 4
TTC |

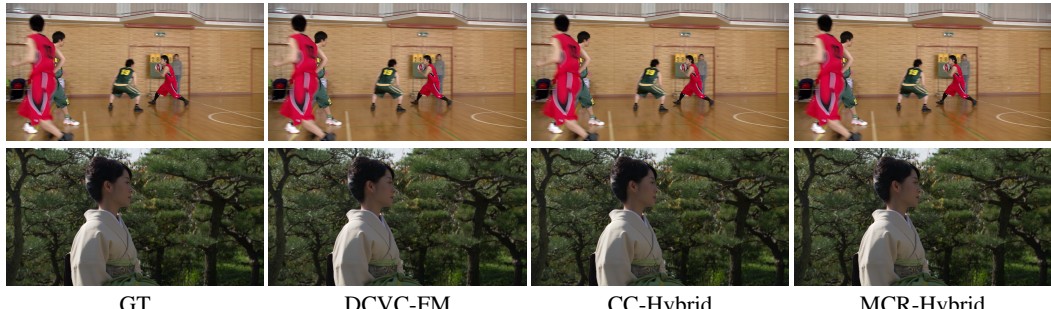

| GT | DCVC-FM | CC-Hybrid | MCR-Hybrid |
|---|---|---|---|

Figure 18: Visualization results.

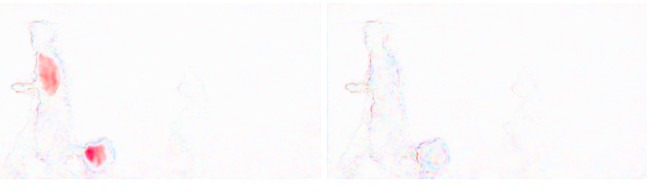

(a) MCR-Hybrid W8A10.        (b) MCR-Hybrid FP32.

Figure 19: We visualize the absolute differences (i.e. the absolute motion errors) between the input and output of the motion codec. The quantization of the motion codec with W8A10 introduces larger motion errors than its FP32 variant without any quantization.

Table 22: List of assets used in the paper with their corresponding license.

| Assets | Licenses |
|---|---|
| Vimeo90K (Xue et al., 2019) | MIT license |
| Compressai (Bégaint et al., 2020) | BSD-3-Clause-Clear License |

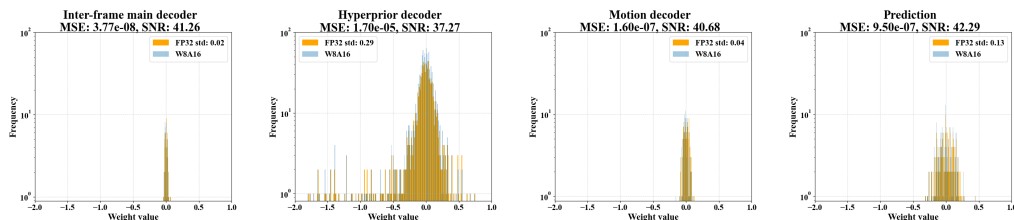

Figure 20: Model weight histogram comparison between FP32 and W8A16 quantization.

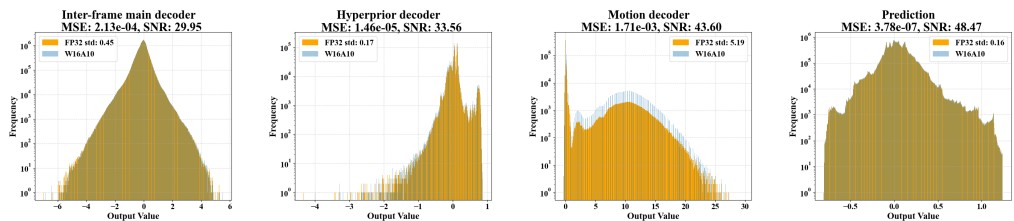

Figure 21: Model activation histogram comparison between FP32 and W16A10 quantization.

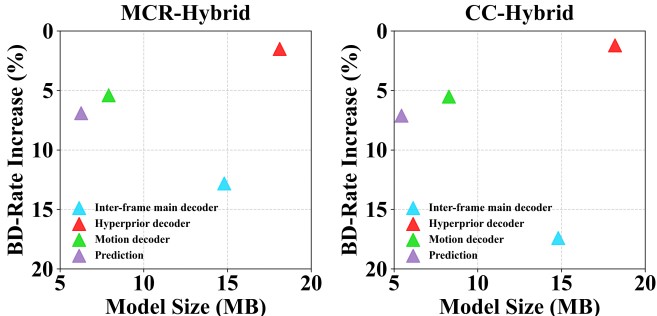

Figure 22: Component-wise analysis of BD-rate increase (due to quantization) vs. model size.

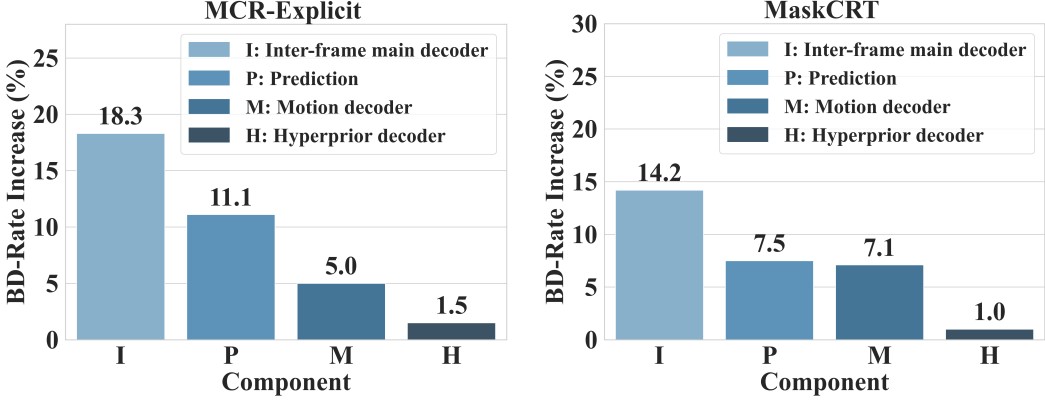

Figure 23: Visualization of the BD-rate increases due to quantization. The component-wise analysis of quantizing MCR-Explicit and MaskCRT to W8A10. The BD-rate increases are measured with the FP32 counterparts serving as anchors.

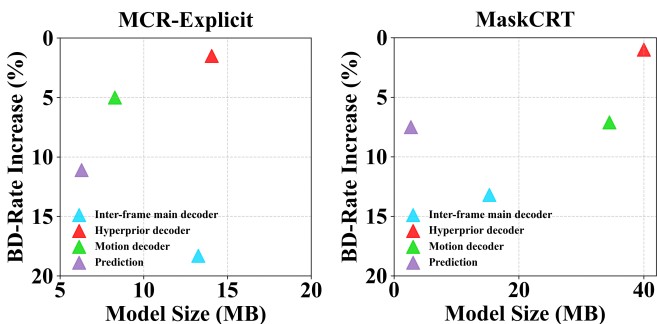

Figure 24: Component-wise analysis of BD-rate increase (due to quantization) vs. model size.

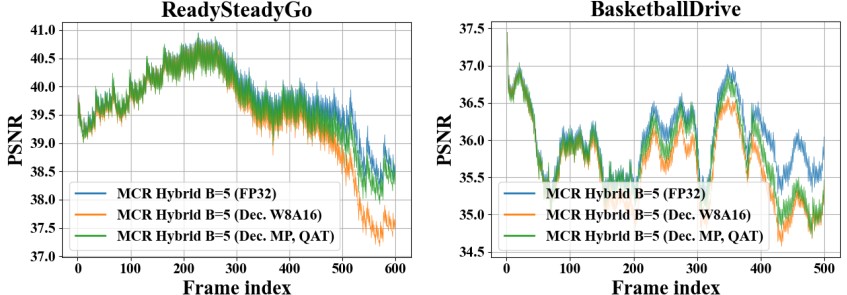

Figure 25: Per-frame PSNR for ReadySteadyGo and BasketballDrive.

