# OpenReview forum: "On the Quantization of Neural Video Codecs"
_ICLR.cc/2026/Conference — Submitted to ICLR 2026_

### Official Review · Reviewer_5r36 · 2025-10-30

**Soundness:** 3
**Presentation:** 1
**Contribution:** 2
**Rating:** 2
**Confidence:** 5

**Summary:**

This paper presents the first systematic study on quantizing neural video codecs across multiple coding frameworks and temporal buffering strategies, introducing a mixed-precision scheme that slashes complexity by 53–87 % with only 2–4 % BD-rate loss.

**Strengths:**

1. The paper systematically investigates quantization in neural video codecs across diverse frameworks and buffering strategies. The experiments comprehensively cover component-wise analysis, multiple quantization methods, and hybrid buffering for fair evaluation.
2. The mixed-precision quantization scheme achieves up to 87 % bit-operation and 53 % model-size savings with only a 3.5 % BD-rate penalty, clearly demonstrating a practical performance–complexity trade-off.
3. The study ties codec quantization to cross-platform reproducibility and temporal drift prevention, providing engineering insights that bridge algorithmic design and system implementation.

**Weaknesses:**

1. The presentation is poor. The paper even lacks an “Abstract” heading on the first page, which is a serious formatting oversight. In addition, there are other presentation issues such as inconsistent font sizes in tables and uneven spacing around some section headings.
2. The quantization evaluation is performed in a floating-point simulation environment without hardware validation. While standard in algorithmic studies, this setup may not capture integer rounding and overflow behaviors on real accelerators, which are crucial for verifying cross-platform consistency and deployment efficiency.
3. The study focuses exclusively on the DCVC-FM backbone, leaving uncertainty about transferability to transformer-based or flow-matching codecs.
4. The paper reports theoretical complexity reductions (bit operations, memory, and model size) but does not include real-device latency or energy measurements. This limits the strength of its claims about practical efficiency.
5. The paper does not quantify how quantization-induced errors accumulate across long temporal dependencies. While short-sequence BD-rate analysis is provided, a study on long-term drift or stability would strengthen claims of robustness in real-time deployment.

**Questions:**

1. Have the authors analyzed temporal error accumulation theoretically or through long-term video sequences to validate stability?
2. How does the quantization behavior vary when applied to different backbone architectures such as DVC or FVC?
3. What are the actual runtime and power benefits on mobile or edge devices, beyond bit-operation and model-size reductions?
4. Have you considered using an automated bit-width allocation or mixed-precision search method (e.g., gradient-based or reinforcement learning approaches) to replace the current manually tuned configuration for better scalability?

---

> ### Author Response · Authors · 2025-11-28
> **Rebuttal to Reviewer 5r36, Part 1**
>
> >**[W1] Editorial comments**
>
> We thank the reviewer for pointing out these formatting issues. We have fixed them in our revised manuscript.
>
>
> ---
>
>
> > **[W2] Cross-platform consistency**
>
> We acknowledge that our work is an algorithmic investigation of quantization effects across neural video coding frameworks and temporal buffering strategies. To derive systematic and holistic insights into their interactions, we follow the established practice in prior studies such as QLIC [R1] and MP-PTQ [R2], conducting quantization in simulated settings. We acknowledge that such simulations are not ideal for validating cross-platform interoperability or for deployment on real hardware. In the revised manuscript (L275), we have explicitly identified these as current limitations and outlined them as directions for future work.
>
>
> ---
>
> > **[W3, Q2] Transferability to other backbone architectures**
>
> To address the reviewer’s concern, we provide a component-wise quantization sensitivity analysis for the Transformer-based codec MaskCRT (Chen et al., 2024b). MaskCRT follows the masked conditional residual coding scheme with an explicit buffer (MCR-Explicit). Similar to the MCR-Explicit with DCVC-FM backbones, MaskCRT exhibits a similar trend in BD-rate increases, reinforcing our claims regarding the relative component-wise sensitivity to quantization in our work. Specifically, the ordering from most to least sensitive is: Inter-frame main decoder, Prediction, Motion decoder, Hyperprior decoder.
>
> To further address the reviewer’s curiosity about transferability, we also present a comparison between each component’s model size and its corresponding BD-rate increase under 8-bit weight and 10-bit activation quantization for both MaskCRT and our masked conditional residual coding framework with explicit buffers (MCR-Explicit). Although the two codecs differ in complexity, there is still a degree of consistency between model size and BD-rate increase. In particular, the hyperprior decoder, despite having the largest model size, experiences only minimal BD-rate degradation under quantization, whereas components with much smaller model sizes exhibit substantially higher performance degradation. This demonstrates that our claims about quantization effects hold across different codec backbones.
>
> Although the high-level coding frameworks such as conditional coding (CC) or masked conditional residual coding (MCR) are generally compatible with various network backbones (e.g., Transformer-based or CNN-based), we believe that using the state-of-the-art implementation from the representative DCVC-FM provides more meaningful insights for future codec development. Drawing conclusions from suboptimal or poorly performing component designs would not be constructive. We believe that our current choice offers the most relevant perspective for understanding how quantization affects modern video coding frameworks and their temporal buffering strategies. We have added these additional results to the Appendix (Section 6.16) to clarify the transferability of our study.
>
> | Component |              MCR-Explicit              |                 |              MaskCRT             |             |
> |-----------|----------------------------------------|-----------------|----------------------------------|-------------|
> |           | Model Size (MB) | BD-Rate Increase (%) | Model Size (MB) | BD-Rate Increase (%) |
> | Inter-frame main decoder | 13.3 | 18.3 | 13.2 | 15.3 |
> | Hyperprior decoder       | 14.1 | 1.5  | 1.0  | 40.0 |
> | Motion decoder           | 8.3  | 5.0  | 7.1  | 34.5 |
> | Prediction               | 6.3  | 11.1 | 7.5  | 2.7  |

---

> ### Author Response · Authors · 2025-11-28
> **Rebuttal to Reviewer 5r36, Part 2**
>
> > **[W4, Q3] Real-device latency and energy measurement**
>
> We clarify that MACs, buffer size, and Decoder BO are inherent and intrinsic to the algorithm design, therefore being independent of the underlying compute platform. This enables a fairer assessment of competing methods, without bias from platform choice or implementation quality.
>
> At this stage, our study remains a systematic algorithmic investigation of quantization effects across neural video coding frameworks and temporal buffering strategies. Our quantization is performed under simulated settings. These simulations do not implement actual fixed-point operations; therefore, we do not report actual runtime and power consumption. However, identifying which modules in neural video codecs are less sensitive to precision reduction in different scenarios, our work can help guide future research toward low-power on-chip neural video codecs. According to Horowitz [R3], floating-point operations require substantially more energy per operation than integer/fixed-point ones. In addition, the energy cost scales approximately linearly with bit-width for additions and roughly quadratically for multiplications.
>
> We further note that real-device latency (or energy consumption) depends heavily on operational complexity, such as the compute overhead needed for function calls and data transfers, as shown in DCVC-RT (Jia et al., 2025). Operational complexity varies significantly across compute platforms. As such, optimization techniques designed to reduce function calls and/or memory transfers on one hardware platform (e.g. from Qualcomm) may not yield the same complexity or runtime benefits on another hardware platform (e.g. from MediaTek). Our work is intended to provide insights from a systematic and holistic perspective that can inform future codec development, instead of optimizing various coding frameworks and buffer strategies on specific hardware platforms.
>
> In the revised manuscript (L273), we have (1) clarified the current limitations of our work in Section 3.3, (2) stated that our focus is on intrinsic and platform-independent complexity measurements, not platform-dependent operational complexity as in (Jia et al., 2025), and (3) indicated that optimizing these variants for specific hardware platforms to assess their performance on real hardware is our future work.

---

> ### Author Response · Authors · 2025-11-28
> **Rebuttal to Reviewer 5r36, Part 3**
>
> > **[W5, Q1]  Accumulated quantization-induced errors**
>
> To address the reviewer’s concern, we report the quantization-induced errors in terms of BD-rate increases over full sequences (intra-period = -1) (Table 8 in the revised manuscript). The results follow trends that are mostly consistent with those presented in the main paper, which report results for 96-frame sequences. Here are our observations regarding the common trends in quantization-induced errors along the temporal dimension for both full and 96-frame sequences:
>
> (1) The coding performance of quantized codecs is inferior to that of FP32 codecs, as accumulated quantization errors gradually degrade frame quality over time. The degradation becomes more prominent on full sequences than 96-frame sequences.
>
> (2) Mixed-precision quantization (Dec. MP, QAT) continues to deliver better coding performance than fixed-precision quantization with 8 bits for weights and 16 bits for activations (Dec. W8A16) while maintaining comparable complexity (L517-527).
>
> (3) In terms of coding performance (BD-rate savings), MCR-Hybrid B=5 (FP32) > MCR-Hybrid B=5 (Dec. MP, QAT) > DCVC-FM (FP32) for both full and 96-frame sequences.
>
> Additionally, we present the temporal degradation in PSNR to illustrate how quantization affects frame quality over time in the Appendix (Section 6.17). It is seen that mixed-precision quantization with quantization-aware training is more effective than fixed-precision quantization.
>
> To further improve clarity, we have (1) included the full-sequence evaluation results in the revised manuscript (Table 8), and (2) provided the PSNR degradation over time in the Appendix (Section 6.17).
>
> *BD-rate (%) comparison for UVG and HEVC-B. All frames are tested under intra-period -1 using BT.601 for the YUV-to-RGB color conversion.*
>
> | **Methods**                        | **UVG** | **HEVC-B** | **Average** |
> |------------------------------------|--------:|-----------:|------------:|
> | DCVC-FM (FP32)                     |   0.0   |     0.0    |     0.0     |
> | DCVC-FM (Dec. W8A16)               |  20.3   |    11.1    |    15.7     |
> | MCR-Hybrid B=5 (FP32)              |  -2.5   |   -10.2    |    -6.35    |
> | MCR-Hybrid B=5 (Dec. W8A16)        |   7.2   |    -0.4    |     3.4     |
> | MCR-Hybrid B=5 (Dec. MP, PTQ)      |  10.2   |    -3.9    |     3.15    |
> | MCR-Hybrid B=5 (Dec. MP, QAT)      |   0.9   |     -7     |    -3.05    |
>
> *BD-rate (\%) comparison for UVG and HEVC-B (same as Table 7 in the main paper). 96 frames are tested under intra-period 32 using BT.601 for the YUV-to-RGB color conversion.*
>
> | **Methods**                        | **UVG** | **HEVC-B** | **Average** |
> |------------------------------------|--------:|-----------:|------------:|
> | DCVC-FM (FP32)                     |   0.0   |     0.0    |     0.0     |
> | DCVC-FM (Dec. W8A16)               |  13.9   |     6.4    |    10.2     |
> | MCR-Hybrid B=5 (FP32)              | -11.7   |   -14.2    |   -13.0     |
> | MCR-Hybrid B=5 (Dec. W8A16)        |  -2.0   |    -8.2    |    -5.1     |
> | MCR-Hybrid B=5 (Dec. MP, PTQ)      |  -5.8   |   -11.1    |    -8.5     |
> | MCR-Hybrid B=5 (Dec. MP, QAT)      |  -8.3   |   -11.5    |    -9.9     |

---

> ### Author Response · Authors · 2025-11-28
> **Rebuttal to Reviewer 5r36, Part 4**
>
> > **[Q4] Automated mixed precision method**
>
> Yes, we recognize that leveraging reinforcement learning to determine the quantization precision of individual decoder components represents a promising direction. For example, the work in [R4] formulates the hyperparameter selection for model training as a reinforcement learning problem. By analogy, the quantization precision assigned to individual decoder components can be viewed as a set of hyperparameters, and the method in [R4] can be readily extended to enable automated mixed-precision quantization. We acknowledge that there are other approaches that could facilitate mixed-precision search. In this work, however, our main focus is to analyze quantization effects across different coding frameworks and buffering strategies, with the mixed‑precision example serving as an illustrative case that highlights the potential of this direction.
>
> We have updated our manuscript (L465-468) to provide a clearer presentation of our contributions and to state that automated mixed-precision quantization is a promising direction for future work.
>
> *Precision settings for W8A16 and mixed-precision (MP).*
>
> | **Precision** | **Components** | **Inter-frame main decoder** | **Prediction** | **Motion decoder** | **Hyperprior decoder** | **Buffer** |
> |---------------|----------------|------------------------------|----------------|----------------------|--------------------------|------------|
> | **W8A16**     | W-bits         | 8  | 8  | 8  | 8  | -  |
> |               | A-bits         | 16 | 16 | 16 | 16 | 16 |
> | **MP**        | W-bits         | 10 | 10 | 10 | 8  | -  |
> |               | A-bits         | 14 | 12 | 14 | 12 | 8  |
>
>
> ---
>
> **Reference**
>
> [R1] H. Sun, L. Yu, and J. Katto, “Q-LIC: Quantizing Learned Image Compression with Channel Splitting,” arXiv preprint arXiv:2205.14510, 2022.
>
> [R2] J. Yu, S. Mai, P. Zhang, Y. Jiang, and J. Cheng, “Mixed-Precision Post-Training Quantization for Learned Image Compression,” IEEE Internet of Things Journal, vol. 12, no. 16, pp. 34392–34405, 2025
>
> [R3] M. Horowitz, “1.1 computing’s energy problem (and what we can do about it),” in 2014 IEEE International Solid-State Circuits Conference Digest of Technical Papers (ISSCC), 2014.
>
> [R4] Y. H. Hung, K.-J. Lin, Y.-H. Lin, C.-Y. Wang, C. Sun, and P.-C. Hsieh, “BOFormer: Learning to Solve Multi-Objective Bayesian Optimization via Non-Markovian RL,” Proc. ICLR, 2025.

---

### Official Review · Reviewer_ELjB · 2025-10-31

**Soundness:** 2
**Presentation:** 3
**Contribution:** 2
**Rating:** 4
**Confidence:** 4

**Summary:**

This paper presents a detailed analysis of the effects of quantization on video compression models, taking into account the following factors: different coding frameworks (e.g., CC, MRC), temporal buffering strategies (Explicit, Implicit, Hybrid), quantization training  strategies (PTQ, QAT), and each decoder component. Based on this, the authors proposed a mixed-precision compression model built upon the MCR architecture, achieving a good rate–distortion–complexity trade-off.

**Strengths:**

1. The RD performance improvement of the proposed model MCR-Hybrid is obvious, as shown in Table 7.
2. The experiments in Section 4 demonstrate a substantial amount of work and provide insightful implications for the design of quantized models.

**Weaknesses:**

The article lacks some key experiments to demonstrate the model's effectiveness.
1. As cross-platform consistency is the most important functionality of INT-quantized compression models, the paper lacks validation in this aspect. Specifically, it remains unclear whether encoding and decoding on different GPU models would introduce reconstruction errors.
2. The comparison of actual encoding and decoding speed is missing in Table 7. Given that it is well known that Decoder BO, including MACs, sometimes fails to reflect the real-time computational cost [r1], such a comparison is necessary.
3. The analysis of decoder components in Section 4.2 lacks reference to complexity metrics. For Conclusion 1, could the inter-frame main decoder suffer the most from quantization simply because it has the highest complexity?

[r1] Zhaoyang Jia, Bin Li, Jiahao Li, Wenxuan Xie, Linfeng Qi, Houqiang Li, and Yan Lu. Towards practical real-time neural video compression, 2025.

**Questions:**

1. In the experiments, how are non-8-bit integer quantizations such as W14/W12/W10 and A14/A12/A10 actually implemented? Do they actually facilitate hardware deployment and deliver real speed-ups?
2. Are the training procedures described in Tables 18–20 complete? DCVC-FM is trained using sequences of up to 32 frames, whereas the authors only adopt a training strategy with up to 5 frames and achieve comparable performance. Can this shortened frame training strategy reproduce the performance of DCVC-FM?

---

> ### Author Response · Authors · 2025-11-28
> **Rebuttal to Reviewer ELjB, Part 1**
>
> > **[W1] Cross-platform consistency**
>
> We clarify that our work is an algorithmic investigation of quantization effects across neural video coding frameworks and temporal buffering strategies. To derive systematic and holistic insights into their interactions, we follow the established practice in prior studies such as QLIC [R1] and MP-PTQ [R2], conducting quantization in simulated settings. We acknowledge, however, that such simulations are not ideal for validating cross-platform interoperability or for deployment on real hardware. In the revised manuscript (L269), we have explicitly identified these as current limitations and outlined them as directions for future work.
>
>
> ---
>
>
> > **[W2] Real-time computational cost**
>
> We clarify that MACs, Buffer size, and Decoder BO are inherent and intrinsic to the algorithm design, therefore being independent of the underlying compute platform. This enables a fairer assessment of competing methods, without bias from platform choice or implementation quality.
>
> At this stage, our study remains a systematic algorithmic investigation of quantization effects across neural video coding frameworks and temporal buffering strategies. Our quantization is performed under simulated settings. These simulations do not implement actual fixed-point operations; therefore, we do not report actual runtime and power consumption. However, identifying which modules in neural video codecs are less sensitive to precision reduction in different scenarios, our work can help guide future research toward low-power on-chip neural video codecs. According to Horowitz [R3], floating-point operations require substantially more energy per operation than integer/fixed-point ones. In addition, the energy cost scales approximately linearly with bit-width for additions and roughly quadratically for multiplications.
>
> We further note that real-device latency (or energy consumption) depends heavily on operational complexity, such as the compute overhead needed for function calls and data transfers, as shown in (Jia et al., 2025). Operational complexity varies significantly across compute platforms. As such, optimization techniques designed to reduce function calls and/or memory transfers on one hardware platform (e.g. from Qualcomm) may not yield the same complexity or runtime benefits on another hardware platform (e.g. from MediaTek). Our work is intended to provide insights from a systematic and holistic perspective that can inform future codec development, instead of optimizing various coding frameworks and buffer strategies on specific hardware platforms.
>
> In the revised manuscript (L273), we have (1) clarified the current limitations of our work in Section 3.3, (2) stated that our focus is on intrinsic and platform-independent complexity measurements, not platform-dependent operational complexity as in (Jia et al., 2025), and (3) indicated that optimizing these variants for specific hardware platforms to assess their performance on real hardware is our future work.
>
> ---
>
> **Reference**
>
> [R1] H. Sun, L. Yu, and J. Katto, “Q-LIC: Quantizing Learned Image Compression with Channel Splitting,” arXiv preprint arXiv:2205.14510, 2022.
>
> [R2] J. Yu, S. Mai, P. Zhang, Y. Jiang, and J. Cheng, “Mixed-Precision Post-Training Quantization for Learned Image Compression,” IEEE Internet of Things Journal, vol. 12, no. 16, pp. 34392–34405, 2025
>
> [R3] M. Horowitz, “1.1 computing’s energy problem (and what we can do about it),” in 2014 IEEE International Solid-State Circuits Conference Digest of Technical Papers (ISSCC), 2014.

---

> ### Author Response · Authors · 2025-11-28
> **Rebuttal to Reviewer ELjB, Part 2**
>
> > **[W3] Complexity metrics**
>
> Quantization errors are influenced by multiple factors. The model complexity is indeed one of them, as components with deeper architectures can accumulate larger errors. As requested, we provide a comparison between each component’s model size and the corresponding BD-rate increase under 8-bit weight and 10-bit activation quantization. The results show that, although the hyperprior decoder has the largest model size, its BD-rate increase due to quantization remains minimal, whereas components with smaller model sizes exhibit substantial performance degradation under quantization.
>
>
> We have included this analysis in the Appendix (Section 6.15).
>
> | Component |            MCR-Hybrid             |                  |            CC-Hybrid             |                 |
> |-----------|-----------------------------------|------------------|----------------------------------|-----------------|
> |           | Model Size (MB) | BD-Rate Increase (%) | Model Size (MB) | BD-Rate Increase (%) |
> | Inter-frame main decoder | 14.8 | 12.8 | 14.8 | 17.4 |
> | Hyperprior decoder       | 18.1 | 1.5  | 18.2 | 1.2  |
> | Motion decoder           | 7.9  | 5.4  | 8.3  | 5.5  |
> | Prediction               | 6.3  | 6.9  | 5.5  | 7.1  |
>
>
>
> ---
>
>
>
> > **[Q1] Hardware deployment**
>
> In our study, we follow the established practice in prior studies such as QLIC [R1] and MP-PTQ [R2], performing quantization (8-bit and non-8-bit) in simulated settings. The simulations allow us to gain insights quickly into various aspects under study. Since these simulations do not implement actual fixed-point operations, they do not deliver real speed-ups.
>
> In the revised manuscript (L273), we have clarified the current limitations of our work in Section 3.3, and put integer network implementations for hardware deployment as our future work.
>
>
> ---
>
>
> > **[Q2] Training procedure for DCVC-FM**
>
> DCVC-FM does not provide the detailed training recipe. Reproducing its results by training from scratch is challenging. Therefore, to ensure a strong DCVC-FM baseline, we reuse the checkpoint released in the original paper, i.e. the pre-trained model, for post-training quantization (the results in Table 7).  For the  competing methods, we follow the established training recipe in (Chen et al., 2024b), which involves 10-frame training.
>
>
> ---
>
> **Reference**
>
> [R1] H. Sun, L. Yu, and J. Katto, “Q-LIC: Quantizing Learned Image Compression with Channel Splitting,” arXiv preprint arXiv:2205.14510, 2022.
>
> [R2] J. Yu, S. Mai, P. Zhang, Y. Jiang, and J. Cheng, “Mixed-Precision Post-Training Quantization for Learned Image Compression,” IEEE Internet of Things Journal, vol. 12, no. 16, pp. 34392–34405, 2025

---

### Official Review · Reviewer_U8x3 · 2025-11-01

**Soundness:** 2
**Presentation:** 3
**Contribution:** 2
**Rating:** 4
**Confidence:** 3

**Summary:**

This paper addresses the problem of the quantization in encoder-decoder neural video codecs. It seeks to present a systematic evaluation of quantization in neural codecs, which have been less explored than for neural image codecs. This is a problem of practical importance due to the increasing use of this class of models and the need for memory-efficient architectures. The primary contribution of the paper is a large number of ablative experiments determining how different components of neural codecs perform under varying bit-widths under uniform post-training quantization (PTQ) and quantization aware training (QAT). A supplementary contribution is the exploration of mixed-precision quantization schemes, and the analysis of temporal buffering strategies under quantization. While the paper contains a large amount of experimentation it would benefit from improved focus and organization, which would improve the strength of the contribution as a systematic evaluation and allow for the practical impact for the paper to be increased.

**Strengths:**

* The paper addresses an important issue. The practical deployment of neural video codecs is increasing, and there is a need for systematic analysis of memory-efficient components in these architectures. While the use of quantization in neural video codecs is not novel (the authors highlight that decoders have frequently incorporated some quantization as part of the encoder or decoder, L128-140), systematic ablation of these features is rarely performed comprehensively and is a valuable contribution.
* The paper is well presented - with clear language, well-presented tables, and clear figures. A large amount of effort has gone into providing clear visual presentation and complete experimental details - which is appreciated.
* The paper contains a very large number of ablative experiments. These are conducted with a large number of configurations (PTQ, QAT; decoder configurations; min-max and MSE quantization; buffering strategies), across multiple datasets. The reported metrics (BD-Rate, PSNR, bpp) / rate-distortion curves in the Appendix are appropriate for video compression.
* While there are some issues of clarity in the characterization of Implicit / Explicit / Hybrid temporal buffers, and the broad classification of residual coding strategies - examining quantization under different architecture classes is well motivated.

**Weaknesses:**

**Major**
* The main issue in the paper is in terms of contribution and positioning. The paper positions itself primarily as a systematic study of quantization effects on neural video codecs, but also introduces some modifications based on mixed-precision quantization and temporal consistency (e.g. pg 5, L239-240, 'a hybrid approach that merges across explicit and implicit buffering strategies'). The paper may be better positioned by focusing the contribution on one aspect in detail (either a proposed new method or a systematic survey). This may additionally help focus the experiments, which while numerous are inconsistent with the contribution scope of a systematic study (e.g. L252, "non-uniform quantization falls outside our scope", and L255 "... generalizing our findings to the full spectrum of quantization methods and network architectures ... is not our intent.") or detailed evaluation of a new method.
* The paper would benefit from including qualitative examples (e.g. comparison of the frames between the different codecs). This is common for video compression papers - and would support the qualitative comments in the paper (e.g. L349-350 "quantization errors in the motion decoder lead to motion errors ... increasing the bitrate" would be useful to show visually).
* The quantization literature could be more complete. There are a large number of quantization works, so in order to position this as "the first systematic study of quantization effects on neural video codecs", it would be useful to put the quantization analysis in context. This is important as the quantization described is currently at an engineering level of different methods, rather than providing theoretical motivation or analysis. For example, reviewing the following may be useful to strengthen the theoretical background:
    * Gholami et al. (2021), A Survey of Quantization Methods for Efficient Neural Network Inference, https://arxiv.org/abs/2103.13630
    * Gersho and Gray (1992), Vector Quantization and Signal Compression https://link.springer.com/book/10.1007/978-1-4615-3626-0
    * Jacob et al. (2017), Quantization and Training of Neural Networks for Efficient Integer-Arithmetic-Only Inference, https://arxiv.org/abs/1712.05877

**Minor**
* There is a separate large line of research on the use of implicit neural representations for video compression (e.g. NeRV[1]), with a large amount of work looking specifically at quantization effects (e.g. Shi et al. (2025), On Quantizing Neural Representation for Variable-Rate Video Coding [2]). These are also occasionally referred to as neural video codecs. It may be worth including a sentence clarifying that the author's work exclusively looks at quantization effects for encoder-decoder hyperprior style codecs.
* It would be useful to explicitly describe the weight / activation quantization bits being evaluated in the method section (the results jump into W8A10 [L309], which reads as arbitrary. The section "Sensitivity of Decoder Components to Quantization" [L309-319] feels like some sentences are presented out of order, making it difficult to draw clear conclusions from.
* There are some survey papers which look at quantization of neural video codecs as components of broader analysis which may be useful to refer to (e.g. Gomes et al. (2025), "End-to-End Neural Video Compression: A Review", https://ieeexplore.ieee.org/document/10962175)

[1] Chen et al. (2021), NeRV: Neural Representations for Videos, https://openreview.net/forum?id=BbikqBWZTGB

[2] Shi et al. (2025), On Quantizing Neural Representation for Variable-Rate Video Coding, https://openreview.net/forum?id=44cMlQSreK

**Questions:**

* The characterization of the temporal correlation frameworks (Figure 2 and 3) appears to be a broad classification of existing methods. This makes it difficult to understand which components are proposed as new contributions or analysis for this paper. It would be useful if the authors could clarify this (and how this relates to the additional hybrid approach in Fig 3c).
* It would be useful to analyse the weight and activation histogram distribution before / following quantization to better understand the sensitivity of the different components to quantization. Evaluating the MSE between the quantized / non-quantized distributions can also help understand the relative importance of bit-widths (beyond the performance level evaluation presented in the paper). E.g. [1], [2].
* Table 5 caption - "X denotes the number of channels" (X doesn't appear used within the table).
* There are a large number of 1-2 letter acronyms (RC, CC, MCR, CRC, I, P, M, H, MP) used for tables. Where space permits, it may be beneficial to use the full names to improve readability.

[1] Han et al. (2016), Deep Compression: Compressing Deep Neural Networks With Pruning, Trained Quantization and Huffman Coding https://arxiv.org/pdf/1510.00149 (Figure 4)

[2] Zhao et al. (2019), Improving Neural Network Quantization without Retraining using Outlier Channel Splitting, https://arxiv.org/pdf/1901.09504 (Figure 1)

---

> ### Author Response · Authors · 2025-11-28
> **Rebuttal to Reviewer U8x3, Part 1**
>
> > **[W1]**  **Contribution and Position**
>
> We thank the reviewer for the suggestion.
>
> First, we emphasize that our work constitutes a systematic investigation of quantization effects across diverse neural video coding frameworks and temporal buffering strategies. Our objective is to generate actionable insights that can guide the future development of neural video codecs.
>
> Second, the hybrid temporal buffering strategy is from [R1]. We have re-phrased our statement (L248-250) and cite [R1] in the revised manuscript to avoid confusion. We note that how it works with diverse coding frameworks and interacts with network quantization is first reported in this work.
>
> Third, mixed-precision quantization introduces no algorithmic modifications to the codec design, apart from applying mixed-precision quantization to individual components. It serves as a direct application of our findings, enabling a more effective balance between coding performance and computational complexity. It reinforces the validity of our analysis and lays the groundwork for more aggressive network quantization.
>
> Fourth, the scope of this study intentionally excludes certain aspects—such as non-uniform quantization and various network design choices for individual components. This is because non-uniform network quantization currently lacks broad hardware support. Furthermore, with limited compute resources, we follow the component designs of DCVC-FM, the SOTA learned neural video codec. We believe drawing conclusions from suboptimal or poorly performing component designs is NOT informative. Our current approach offers the most meaningful perspective for understanding how quantization effects influence modern video coding frameworks and their temporal buffering strategies.
>
> To highlight the connections between our contributions and the supporting experiments, we restate the contribution statements from the Introduction here for ease of reference.
>
> * (A) We analyze multiple neural video coding frameworks and temporal buffering strategies to examine how quantization affects their coding performance across a wide range of bit-widths (L107);
>
> * (B) We conduct extensive analyses to assess how each decoding component responds to quantization effects (L114);
>
> * (C) we explore a mixed-precision quantization scheme for various coding variants to strike a balance between coding efficiency and computational complexity (L115).
>
> Our supporting experiments are as follows:
>
> * Section 4.2 evaluates contribution (A) by examining the coding performance of two common coding frameworks - conditional coding and masked conditional residual coding - under quantization.
>
> * Section 4.2 supports (B) by decomposing the decoder into individual components and varying the bit-width of each component independently.
>
> * Section 4.3 examines contribution (A) by evaluating different temporal buffering strategies under quantization, including explicit, implicit, and hybrid buffers.
>
> * Section 4.4 addresses (C) by comparing mixed-precision quantization with uniform quantization for state-of-the-art codecs.
>
> To further enhance clarity, we have (1) included the corresponding section numbers directly after each contribution in the revised manuscript (L107-117), and (2) re-phrased our statement (L248-250) and cite [R1] in the revised manuscript.
>
>
> ---
>
>
> > **[W2] Qualitative results**
>
> Thank you for the suggestion. In the Appendix of our revised manuscript, we have additionally provided visualizations comparing the reconstructed video frames generated by different codecs (Fig. 18 in the revised manuscript). Moreover, we also visualize motion errors due to codec quantization (Fig. 19 in the revised manuscript).
>
>
> ---
>
>
> > **[W3] Quantization literature**
>
>
> We thank the reviewer for the suggestion. In our revised manuscript, we have added citations to (Gholami et al., 2021) and (Jacob et al., 2017) in the Introduction (L49). These studies present fundamental quantization concepts for neural networks, many of which are also discussed in the white paper (Nagel et al., 2021) cited in our initial manuscript. By contrast, Gersho and Gray (1992) focus on classical quantization techniques for general signal compression, which are less relevant to our scope - specifically, the quantization of neural video codecs.
>
> ---
>
> **Reference**
>
> [R1] Y.-H. Chen, Y.-C. Yao, K.-W. Ho, C.-H. Wu, H.-T. Phung, M. Benjak, J. Ostermann, and W.-H. Peng, “HyTIP: Hybrid Temporal Information Propagation for Masked Conditional Residual Video Coding,” in Proc. IEEE/CVF Int. Conf. Comput. Vis. (ICCV), 2025.

---

> ### Author Response · Authors · 2025-11-28
> **Rebuttal to Reviewer U8x3, Part 2**
>
> > **[MW1] Implicit Neural Representations**
>
> We thank the reviewer for the suggestion. In our revised manuscript, we have added citations to these prior works and clarified that implicit neural representation (INR)-based learned video codecs, while employing network quantization in an overfitted setting to represent a video in compressed form, are an orthogonal line of research distinct from our central focus on variational autoencoder (VAE)-based codecs (L262).
>
>
> ---
>
>
> > **[MW2] Weight / activation quantization clarification**
>
> We thank the reviewer for the valuable feedback. In our experiments, we first determine the precision at which the decoder begins to collapse in coding performance and then use this precision as the starting point for our analysis across different video codecs. Following the suggestion, we have further clarified in the “The Sensitivity of Decoder Components to Quantization” section to better articulate the goals of our experiments (L350-L361) in the revised manuscript.
>
>
> ---
>
>
> > **[MW3] Quantization survey**
>
> We thank the reviewer for the suggestion. We have included Gomes et al. (2025) in the Introduction (L96). This paper provides a high-level overview of end-to-end neural video codecs but does not explicitly address the quantization of neural video codecs, which remains an emerging research area. Our work constitutes the first systematic and comprehensive study of quantization effects in neural video codecs.
>
>
> ---
>
>
> > **[Q1] Coding frameworks and temporal buffers clarification**
>
> We thank the reviewer for the suggestion. We clarify that Figure 2 presents a classification of existing neural video coding frameworks, while Figure 3 classifies existing temporal buffering strategies. None of these frameworks or strategies are newly proposed in this work. Rather, we emphasize that our comprehensive analysis investigates how activation and weight quantization affect the trade-off between coding performance and computational complexity across various combinations of these frameworks and buffering strategies. The attempt at analyzing the interaction between neural video coding frameworks and temporal buffering strategies in the presence of quantization constitutes our unique and novel contributions. Our work sheds light on future codec development. Notably, the quantization of neural video codecs remains an emerging yet important research topic. Our work approaches this problem from a holistic perspective. In contrast, some early works, such as MobileNVC (van Rozendaal et al., 2024)  and DCVC-RT (Jia et al., 2025), report quantization results exclusively for specific coding frameworks and do not consider their interaction with various temporal buffering strategies (see Table 1).
>
> In our revised manuscript, we have made these clear in (L253-256).
>
>
> ---
>
> > **[Q2] Weight and activation distribution analysis**
>
> We thank the reviewer for the suggestion. In our work, however, analyzing the weight and activation histogram distributions alone is not sufficient to draw conclusions about the quantization sensitivity of different components. First, our work adopts per-tensor quantization for activations and per-channel quantization for network weights, techniques that are well discussed in (Nagel et al., 2021). Because per-tensor quantization assigns a distinct quantizer to each layer output, it becomes difficult to determine which layer should be examined for each decoder component—especially since each component consists of many blocks and layers, making it impractical to visualize all activations. A similar challenge arises for weights: per-channel quantization provides a finer granularity than per-tensor quantization, giving the convolutional kernel for each output channel its own quantizer. Consequently, identifying which output channels are most critical for visualization is also non-trivial. Furthermore, histogram analysis of individual weights or activations does not account for the interactions among the codec’s components, which have a substantial impact on overall coding performance.
>
> To address the reviewer’s curiosity, we visualize the histograms of the weight channel and the activation tensor with the highest variances for each decoder component under two quantization settings-W8A16 for network weights and W16A10 for activations in MCR-Hybrid. These results can found in the Appendix (Section 6.14). These two settings help isolate the quantization effects on either weights or activations. As said, the results are inconclusive and may be even misleading. For example, in Figure 21, one may be tempted to believe that the hyperprior decoder should be assigned high precision as their signal-to-noise-ratios due to the same activation quantization (W16A10) are relatively low. As a matter of fact, our BD-rate analysis indicates that this component is less critical.
>
> ---
>
> > **[Q3, Q4] Editorial comments**
>
> We thank the reviewer for the suggestions. We have fixed them in our revised manuscript.

---

### Author Response · Authors · 2025-12-03
**Final Remarks by Authors**

We thank all the reviewers for carefully reviewing our manuscript and providing constructive feedback. Notably, they acknowledged the following merits of our work:
* Our work addresses a practical issue in neural video codec development by analyzing how coding frameworks and temporal buffering strategies interact under quantization constraints, offering novel insights that bridge algorithmic design and system implementation. **(U8x3, ELjB, 5r36)**
* Our study is both motivated and essential, given the demand for the deployment of neural video codecs under strict memory-access constraints and across compute platforms. **(U8x3)**
* We provide comprehensive experiments to evaluate various aspects of different coding frameworks under quantization, including component-wise analysis, multiple quantization methods, and buffering strategies. **(U8x3, ELjB, 5r36)**
* The investigated mixed-precision scheme demonstrates a practical performance–complexity trade-off compared with the uniform quantization scheme. **(ELjB, 5r36)**


The major concerns from the reviewers are addressed carefully in our revised manuscript and rebuttal responses:
* **Hardware validation:** We follow the common practice in many quantization studies [R1, R2, R3, R4], using simulations and platform-independent metrics to examine the effects of quantization on neural video codecs. We emphasize that our work is the first comprehensive study that addresses quantization effects across various coding frameworks and buffering strategies, with the aim of generating actionable insights into the development and deployment of future neural video codecs.
* **Generalizability:** By including additional results for the transformer-based codec in the Appendix (Section 6.16), we validate the generalizability of our claims regarding the relative component-wise sensitivity to quantization in other neural video codec designs.

We have revised our manuscript to improve clarity and better convey the contributions, scope, and limitations of our work. We hope that the revised version clearly communicates the merits of our study.

Overall, our work represents the first systematic attempt to investigate quantization effects across modern coding frameworks and temporal buffering strategies, laying a foundation for future research in this direction. Although several reviewers noted the absence of hardware validation, this does not diminish the value of quantization simulation for understanding quantization behaviors in neural video codecs. As stated, our goal is to provide systematic and holistic insights that can inform future codec development. While optimizing various coding frameworks and buffering strategies for specific hardware platforms is valuable future work, it is not the focus of our current study. Moreover, all reviewers **(U8x3, ELjB, 5r36)** recognized the significance of our contributions to the future development of neural video codecs, as well as the comprehensive and informative experiments we conducted to bridge algorithmic design with system-level implementation. We hope the area chairs and reviewers recognize that our work offers timely and significant insights into the practical aspects of neural video codecs, and look forward to receiving a favorable final decision.

**References:**

[R1] J. Y. Yang, B. Kim, J. Bae, B. Kwon, G. Park, E. Yang, S. J. Kwon, and D. Lee, ”No Token Left Behind: Reliable KV Cache Compression via Importance-Aware Mixed Precision Quantization,” arXiv preprint arXiv:2402.18096,2024.

[R2] Y. Bondarenko, R. Del Chiaro, and M. Nagel, ”Low-Rank Quantization-Aware Training for LLMs,” arXiv preprint arXiv:2406.06385, 2024.

[R3] H. Sun, L. Yu, and J. Katto, “Q-LIC: Quantizing Learned Image Compression with Channel Splitting,” arXiv preprint arXiv:2205.14510, 2022.

[R4] J. Yu, S. Mai, P. Zhang, Y. Jiang, and J. Cheng, “Mixed-Precision Post-Training Quantization for Learned Image Compression,” IEEE Internet of Things Journal, vol. 12, no. 16, pp. 34392–34405, 2025

---

### Meta-Review · Area_Chair_ECE9 · 2026-01-07

**Summary:**

Reviewers expressed concerns about the level of novelty and contribution of the paper, noting that the analysis of quantization effects in neural video codecs largely builds on established techniques and observations from prior work. Several reviewers questioned whether the proposed insights go beyond careful empirical characterization, or lead to new principles, methods, or actionable guidelines for codec or model design. Additional concerns were raised regarding the limited range of codecs and quantization schemes evaluated, as well as the clarity of the paper’s positioning relative to existing literature.

**Reviewer Concerns:**

The rebuttal clarified the authors’ intent to provide a systematic and diagnostic study rather than a novel algorithmic contribution, and addressed some questions about experimental setup and evaluation metrics. These clarifications improved the presentation and helped resolve minor misunderstandings regarding scope. However, the main concerns regarding limited novelty, incremental contribution, and unclear broader impact remain largely outstanding. In particular, the rebuttal did not convincingly demonstrate how the presented findings substantially advance the state of the art or change current practice in neural video compression. Besides, this paper has a citation hallucination problem--flagged by Program Chairs.

**Reviewer Scores:**

442. With a full post-rebuttal discussion, some reviewers might have slightly increased their scores to reflect improved clarity and framing. However, it is unlikely that these changes would have materially altered the overall assessment or the final recommendation, as the central concerns would likely have persisted.

---

### Decision · Program_Chairs · 2026-01-26

Reject